# Performative Learning Theory

**Julian Rodemann** [1 2]   **Unai Fischer-Abaigar** [2 3]   **James Bailie** [4]   **Krikamol Muandet** [1]

## Abstract

Performative predictions influence the very outcomes they aim to forecast. We study performative predictions that affect a sample (e.g., only existing users of an app) and/or the population (e.g., all potential app users). This raises the question of how well models generalize under performativity. For example, how well can we draw insights about new app users based on existing users when both of them react to the app's predictions? We address such questions by embedding performative predictions into statistical learning theory. Our goal is to initiate the study of learnability under performativity. We prove generalization bounds under performative effects on the sample, on the population, and on both. A key intuition behind our results is that in the worst case, the population negates predictions, while the sample deceptively fulfills them. We cast such self-negating and self-fulfilling predictions as min-max and min-min risk functionals in Wasserstein space, respectively. Our analysis reveals both a fundamental trade-off between performatively changing the world and learning from it, as well as a surprising insight on how to improve generalization guarantees by retraining on performatively distorted samples. We illustrate our bounds using real data on prediction-informed assignments to job trainings.

## 1. Introduction

Machine learning (ML) systems have evolved from merely analyzing the world to shaping it. Their predictions have

[1]Rational Intelligence Lab, CISPA Helmholtz Center for Information Security, Saarbrücken, Germany [2]Department of Statistics, LMU Munich, Germany [3]Munich Center for Machine Learning (MCML), Germany [4]Department of Computer Science and Engineering, Chalmers University of Technology and University of Gothenburg, Sweden. Correspondence to: Julian Rodemann <julian.rodemann@cispa.de>.

*Proceedings of the 43rd International Conference on Machine Learning*, Seoul, South Korea. PMLR 306, 2026. Copyright 2026 by the author(s).

real-world effects on what these systems aim to predict in the first place (Liu et al., 2026). Building on work by Morgenstern (1928); Grunberg & Modigliani (1954) in the social sciences, Perdomo et al. (2020) formalize such feedback loops as *performative* predictions (PP). Examples range from self-fulfilling prophecies in financial markets (Neurath, 1911; Soros, 1994; MacKenzie, 2008) and strategic behavior in selection processes (Khandani et al., 2010; Vo et al., 2024) to self-negating traffic route predictions (Benenati & Grammatico, 2024), gaming recommender systems (Bian et al., 2023) and early warning systems in high schools (Perdomo et al., 2025). We refer to Hardt & Mendler-Dünner (2025) for a recent survey on performative predictions.

In these examples, predictions of outcomes have performative effects on the actual outcomes (e.g., a navigation system predicting congestion may cause drivers to reroute, making the predicted congestion disappear). Both the initial framework (Perdomo et al., 2020) and subsequent work (Miller et al., 2021; Brown et al., 2022; Mofakhami et al., 2023; Perdomo, 2023; 2024; 2025) define these actual outcomes as—statistically speaking—the whole population of units. Since their goal is to study stability and optimality of PP, these works do not address learnability of the population from a sample under performativity.

In the article at hand, we generalize this setting by studying performative effects on a sample drawn from the population, on the population itself, as well as on both. In many applications, only a finite sample of training data is available instead of complete access to the entire population. Performative effects might arise both in and out of this sample, see Sections 1.2 and 4 for real-world examples. In other words, while prior works focus on repeated *risk* minimization under *population* performativity, we turn our attention additionally to repeated *empirical risk* minimization (ERM, Vapnik, 1998) under *empirical* and *population* performativity. This raises the question whether and how well models can generalize from train to test data if the train or test data (or both) are subject to performative effects. We prove bounds on the generalization error, the generalization gap and the excess risk in all these scenarios under generic assumptions on the performative effects. In particular, we do not assume any specific knowledge about the performative effects—besides being continuous as in Perdomo et al. (2020); Brown et al. (2022). This is in contrast to strategic classification (as-

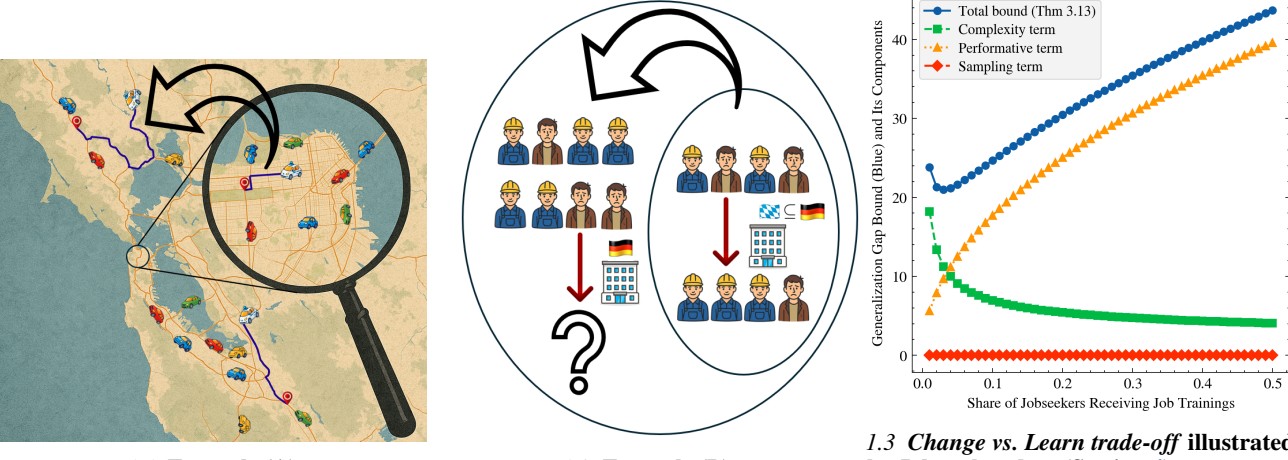

*Figure 1. 1.1* **Example (A):** Route predictions are known to have performative effects: Drivers avoid routes with predicted congestion, thereby rendering these predictions less accurate. Can routing apps still generalize from San Francisco (sample) to the whole Bay Area (population)? *1.2* **Example (B):** A job center in Bavaria assigns job training programs to those among the unemployed that have high risks of long-term unemployment according to machine learning (ML) predictions. As a result of the job training, their probability of finding a job increases, a textbook example of a performative effect, see Section 4. Can the job center's ML model trained on performatively shifted data from Bavaria generalize to the whole German population, which will in turn react to the predictions? *1.3* **Theorem 3.13 applied to job seeker data:** Growth of generalization gap bound (blue) shows trade-off between performatively changing a population (by assigning more people to job trainings) and reliably learning its properties. Details on the bound's components (red, green, yellow) are in Section 4.

suming best-response under explicit costs), causal modeling (structural assumptions), or algorithmic game theory and mechanism design (assuming equilibrium-based response models with rational agents), see discussion of related work in Section 5 and Appendix B.

## 1.1. Summary of Contributions

This paper introduces performative learning theory (PLT): a statistical learning-theoretic framework for studying generalization when predictions change the data-generating process. A necessary first step is to develop a conceptual grasp of what generalization actually means if the sample, population or both performatively react to predictions. Section 2 achieves this, triggering four main research questions **RQ1** through **RQ4**. We formally embed PP into learning theory, allowing us to answer **RQ1–RQ3**. Specifically, we prove generalization bounds on PP, requiring only a compact parameter space and a subset of the assumptions in the original PP framework (Perdomo et al., 2020; Brown et al., 2022). Moreover, we give tighter bounds under additional assumptions on the hypothesis class. We find that two factors complicate generalization under performativity: Not only can the population negate our predictions (self-defeating), the sample can also deceive us by doing the opposite and reinforce them (self-fulfilling), creating an *empirical echo chamber*. Technically, we cast these self-negating and self-fulfilling predictions as min-max and min-min learning problems in Wasserstein space, respectively. This allows us to leverage empirical process theory for dual

characterization of locally-distributionally robust (Gao & Kleywegt, 2023) and favorable (Jiang & Xie, 2025) learning, respectively. Conceptually, our bounds highlight a fundamental trade-off between performatively changing data and drawing reliable conclusions from it. We further provide a corollary that allows for improving generalization bounds by retraining on performatively distorted samples. We illustrate our results in a case study on performative effects of unemployment risk predictions, harnessing administrative labor market records from 1975 to 2017 in Germany (raw data with over 60 million rows) provided by the German Federal Employment Agency.

## 1.2. Running Examples

We introduce two prototypical examples of generalization under performativity. In Example (A), performative effects are induced by the strategic behavior of agents (endogenous), while in Example (B), they result from treatments (exogenous). Our results address both scenarios. Both examples involve a model $\widehat{\theta}_{t+1}$ trained on a sample $\widehat{d}_t$, which was drawn from a population $d_t$. As is customary, we assume $\widehat{d}_0 \overset{\text{iid}}{\sim} d_0$ for the initial sample $\widehat{d}_0$ and the initial population $d_0$. Both quantities can change over time, and hence are indexed by $t \in \{1, \dots, T\}$.

**(A) Routing App:** A company offers mapping services that help users find locations and get directions. Their route planning is known to have performative effects on the users. For instance, drivers avoid routes with predicted

traffic jams, jeopardizing these prediction's accuracy by altering the ground truth—a classic example of *self-negating* performative predictions (Cabannes, 2019; Bagabaldo et al., 2024; Benenati & Grammatico, 2024). Suppose the company now wants to use a new model for the route predictions. In order to test it, they first run the model $\widehat{\theta}_1$ in beta stage on selected users (sample $\widehat{d}_0$), say, only in San Francisco (see Figure 1.1). The performative effects of $\widehat{\theta}_1$ alter this sample $\widehat{d}_0$ and thus the subsequently updated models, giving $\widehat{d}_1, \ldots, \widehat{d}_T$ and $\widehat{\theta}_1, \ldots, \widehat{\theta}_T$. Before releasing the new model to all users $d_0$ in, say, the whole Bay Area (Figure 1.1), the company would like to know the worst case error of their current model $\widehat{\theta}_t \in \{\widehat{\theta}_1, \ldots, \widehat{\theta}_T\}$ on all users $d_0$ (or performatively affected $d_t$). Our bounds allow for nuanced answers to this question. They tell the company the generalization error that their model will not exceed, depending on the sample size $n$ of $\widehat{d}_0$ and on the (observable) number $m$ of drivers in that sample that changed their route due to the predictions, giving rise to what we call the *performative response rate* $m/n$. The latter serves as an estimator of the performative response rate in the whole Bay Area.

**(B) Profiling of Job Seekers (Section 4)**: Many public employment services (PES) attempt to predict the risk of newly registered job seekers becoming long-term unemployed (Allhutter et al., 2020; Kern et al., 2021; Junquera & Kern, 2025). These risk predictions—generated by caseworkers or ML models—determine access to scarce job training programs, since job centers assume high-risk individuals to benefit the most from those trainings (Ernst et al., 2024; Fischer-Abaigar et al., 2025). Thus, the predictions exhibit a performative effect: If the model predicts a high risk of prolonged unemployment for a person, they are more likely to be allocated to a training program, leading to a quicker reintegration in the labor market (if the training is effective). Suppose the German PES tests a new ML model on selected job centers (e.g., only in Bavaria, see Figure 1.2, or in randomly selected job centers) and observes how this model's predictions affect (via assignments to training programs) whether job seekers exit unemployment. Can we help the agency by giving guarantees on their model's generalization error when rolled out to all job centers? Section 4 answers affirmatively by illustrating our generalization bounds on real data from the German PES.

## 2. Generalization Under Performativity

We assume our data live in a compact subset $\mathcal{Z} = \mathcal{Y} \times \mathcal{X}$ of $\mathbb{R}^v$ with $\mathcal{Y} \subset \mathbb{R}^{\nu_y}$, $\mathcal{X} \subset \mathbb{R}^{\nu_x}$ and $\nu = \nu_y + \nu_x$. Random variables on $\mathcal{Z}$ will be denoted by $Z_1, \ldots, Z_n$, probability distributions on $\mathcal{Z}$ by $d$, and the set of all such $d$ by $\Delta$. Let $\mathcal{F} := \{f_\theta : \mathcal{X} \to \mathcal{Y} \mid \theta \in \Theta\}$ be our hypothesis class, whose parameter space $\Theta$ we assume to be a compact and convex subset of Euclidean space (akin to Perdomo

et al., 2020). We equip $\mathcal{X}, \mathcal{Y}, \mathcal{Z}$ and $\Theta$ with the Euclidean ($L_2$) norm $\|\cdot\|_2$, and define $F := \sup_{\theta \in \Theta, x \in \mathcal{X}} \|f_\theta(x)\|_2$. Observe that $F$ is finite because $\mathcal{Y}$ is bounded.

**Definition 2.1** (Risk (Minimizer) and Loss). The *risk* of the parameter $\theta \in \Theta$ on data generated according to the distribution $d \in \Delta$ is defined as $\mathscr{R}(d, \theta) = \mathbb{E}_{Z \sim d}[\ell(Z, \theta)]$, where $\ell : \mathcal{Z} \times \Theta \to \mathbb{R}$ is a *loss function*. For any distribution $d \in \Delta$, we call

$$G(d) = \arg\min_{\theta \in \Theta} \mathscr{R}(d, \theta),$$

the *risk minimizer*, which we assume to be unique.

Given independent and identically distributed (i.i.d.) training data $Z_1, \ldots, Z_n$ drawn from a true but unknown distribution $d_0$, we seek some $\theta \in \Theta$ whose risk $\mathscr{R}(d_0, \theta)$ is close to the true, unknown minimum risk $\inf_{\theta \in \Theta} \mathscr{R}(d_0, \theta)$, often referred to as *Bayes* risk. The risk $\mathscr{R}(d_0, \theta)$ of $\theta$ with respect to the true $d_0$ is known as *generalization error* of $\theta$.

**Definition 2.2** (Excess Risk). The difference $\mathscr{R}(d, \theta) - \inf_{\theta' \in \Theta} \mathscr{R}(d, \theta')$ is called the *excess risk* of $\theta$.

A popular learning strategy is ERM (Vapnik, 1998). Let $\widehat{d} := \frac{1}{n} \sum_{i=1}^{n} \delta_{Z_i}$ be the *empirical distribution* of $Z_1, \ldots, Z_n$, where $\delta_{Z_i}$ is the Dirac measure at $Z_i \in \mathcal{Z}$. We denote the *empirical risk minimizer* $G(\widehat{d})$ by $\widehat{\theta}$. We now introduce the general setting of *performative predictions* (Perdomo et al., 2020), where the deployment of a predictive model influences the distribution of future data. For maximal generality, we adopt the *stateful* extension of performative prediction in which model predictions and the current data distribution jointly induce tomorrow's distribution (Brown et al., 2022). Formally, the evolution of the distribution is governed by an unknown transition map Tr as follows: In each round $t \in \{1, \ldots, T\}$, an institution deploys $\theta_t$ and subsequently observes the induced distribution

$$d_t = \mathrm{Tr}(d_{t-1}, \theta_t, ).$$

Note that the *stateless* $d_t = \mathrm{Tr}_s(\theta_t)$ by Perdomo et al. (2020) is a special case. The *stateful performative risk* of $\theta$ is defined as $\mathbb{E}_{Z \sim d_t}[\ell(Z, \theta)] = \mathbb{E}_{Z \sim \mathrm{Tr}(d_{t-1}, \theta)}[\ell(Z, \theta)]$.

**Definition 2.3** (Repeated (Empirical) Risk Minimization). A natural approach in the performative setting is to update the model through *repeated risk minimization (RRM)*. At each round $t$, the new model parameter $\theta_{t+1} = G(d_t)$ is chosen to minimize the (true) risk on the current distribution $d_t$. As hinted at above, we additionally study *repeated empirical risk minimization (RERM)* which is defined analogously for $Z_1, \ldots, Z_n \sim \widehat{d}_t = \mathrm{Tr}(\widehat{d}_{t-1}, \widehat{\theta}_t)$ as $\widehat{\theta}_{t+1} = G(\widehat{d}_t)$.

Perdomo et al. (2020); Brown et al. (2022) study stability and optimality of $\theta$ in these sequences $\{\theta\}_{t=1}^T$ and $\{\widehat{\theta}\}_{t=1}^T$. We explain how stable $\theta$'s and optimal $\theta$'s are subsumed by our analysis in Appendix A.

*Table 1.* Overview of Research Questions **RQ1**–**RQ4** on generalization under performativity. The setup with no retraining and no performativity is the classical learning theory setup, where ERM is Bayes optimal. Retraining on new samples without performativity is a special case of online learning. The performative prediction (PP) setup established by Perdomo et al. (2020), Brown et al. (2022) and Perdomo (2025) considers performativity on the whole population and retraining of models on the whole population. Perdomo et al. (2020, Sec. 3.4) also considers retraining on data sampled anew (*i.i.d*) in each iteration from the performatively changed population. Dashes ("–") mark nonsensical or inapplicable combinations. For instance, retraining on a sample without performative effects is nonsensical since the sample does not change and so retraining would correspond to doing the exact same thing multiple times. Retraining on both the sample and the population is relevant when we have access to a sample first, and then later to the whole population (sequentially) or for statistical analysis with a hypothetical process on the assumed population in parallel (simultaneously).

| Performative Effects | Retraining | | | | | |
|---|---|---|---|---|---|---|
| | **None** | **On Sample** | | **Population** | **Both,[1] Sequentially** | **Both,[1] Simultaneously** |
| | | **Drawn Once** | **Drawn Anew** | | | |
| **None** | ERM | – | Online Learning | – | – | – |
| **On Sample** | – | **RQ1** | – | PP | – | – |
| **On Population** | **RQ2** (a)[*] | **RQ2** (b) | PP (Sec. 3.4) | PP | **RQ3** (a) | – |
| **On Both** | **RQ2** (c) | **RQ2** (d) | PP (Sec. 3.4) | PP | **RQ3** (b) | **RQ4**[**] |

[1] With the sample being drawn once. [*]Partly answered by Kirev et al. (2025). [**]Asymptotically answered by Li et al. (2025b).

## 2.1. On the Meaning of Performative Generalization

*To what extent can models generalize from a finite sample to the population in a performative world?* We study this question under performativity in the sample, in the population or in both. Interestingly, performative generalization can differ across these scenarios, depending on the data on which the model is (re)trained: the sample, the population, or (sequentially or simultaneously) on both. Table 1 summarizes all of the resulting scenarios, including familiar paradigms like ERM, online learning and classical PP as well as four open research questions (RQs).

(**RQ1**) Can we bound the (classical) *excess risk* (Def. 2.2) $\mathscr{R}(d_0, \widehat{\theta}_t) - \inf_{\theta \in \Theta} \mathscr{R}(d_0, \theta)$ for models $\widehat{\theta}_t$ iteratively retrained on samples $\widehat{d}_{t-1} = \text{Tr}(\widehat{d}_{t-2}, \widehat{\theta}_{t-1})$? In this scenario, a model exhibits performative effects on the sample (or subpopulation) it is trained on, but not on the whole population. As illustration, consider Example (A) in which a navigation app's route predictions are being made visible only to selected users (sample), not to all users (population).

(**RQ2**) Can we bound the *performative excess risk*

$$\mathscr{R}(\text{Tr}(d_0, \widehat{\theta}_t), \widehat{\theta}_t) - \inf_{\theta \in \Theta} \mathscr{R}(\text{Tr}(d_0, \widehat{\theta}_t), \theta)$$

for any $t \in \{1, \dots, T\}$? This is the performative version of the classic excess risk (Def. 2.2). The first term is the *performative generalization error*. It describes the error a model $\widehat{\theta}_t$ (trained on a sample $\widehat{d}_{t-1}$) suffers when being deployed on the population $d_0$ under performativity. In this case, the performative effect of $\widehat{\theta}_t$ will shift $d_0$ to $d_1 = \text{Tr}(d_0, \widehat{\theta}_t)$. The model is then evaluated on this performatively shifted distribution. The second term is the Bayes risk on this shifted $d_1$ within the "hypothesis class" $\Theta$. **RQ2** (a): In

case of no retraining, $\widehat{\theta}_1$ is found via standard ERM. After being deployed on the population, it causes a performative shift $d_1 = \text{Tr}(d_0, \widehat{\theta}_1)$. This special case is partly answered by Kirev et al. (2025) for binary classification under linear performative shift of $Y \mid X$ and marginal $X$. Our results are more general in two ways. First, we account for all Lipschitz continuous transition maps as in Perdomo et al. (2020); Brown et al. (2022). Second, we subsume performative shifts on any subsets of $Y$ and $X$. **RQ2** (b): Retraining on the same $\widehat{d}_1$ (due to the absence of performative effects) gives $\widehat{\theta}_1 = \widehat{\theta}_2 = \cdots = \widehat{\theta}_T$. **RQ2** (c) is subsumed by (a), as performative effects on the sample are irrelevant (do not change $\widehat{\theta}_1$) due to lack of retraining. **RQ2** (d) combines **RQ1** and **RQ2** (a): In addition to (multi-shot) sample performativity $\widehat{d}_t = \text{Tr}(\widehat{d}_{t-1}, \widehat{\theta}_t)$ as in **RQ1**, we have to account for (single-shot) population-level performative shifts $d_1 = \text{Tr}(d_0, \widehat{\theta}_t)$ as in **RQ2** (a). Like the standard excess risk, the performative excess risk is subject to uncertainty about the unknown distribution $d_0$. However, the performative excess risk is also subject to uncertainty about unknown $\text{Tr}(\cdot, \cdot)$. For scenario **RQ2** (d), consider again Example (A): The routing model is (re)trained on selected users (sample), but predictions are made public to all users in $d_0$, causing a performative shift to $d_1$. As we do not retrain on the population, we do not have to account for performative effects beyond the one shifting $d_0$ to $d_1$. This changes in **RQ3**.

(**RQ3**) Can we bound the *cumulative performative excess risk* of a model that was first trained $T$ times on the sample and then $\tilde{T} - T$ times on the population

$$\sum_{t=T}^{\tilde{T}} \left( \mathscr{R}(d_t, \theta_t) - \inf_{\theta \in \Theta} \mathscr{R}(d_t, \theta) \right)$$

with $d_t = \text{Tr}(d_{t-1}, \theta_t)$ for $t \geq T$, $\theta_T = \widehat{\theta}_T$ and $d_{T-1} = d_0$? Note that we cannot disregard the retraining on shifted samples, because this affects the population via the initial performative effect of the deployed $\widehat{\theta}_T$. While **RQ2** asks for the generalization error of a model on a population reacting to the model's predictions *once*, **RQ3** considers the accumulated generalization error on a population reacting to ($\tilde{T} - T$ times) repeatedly retrained models $\theta_t$ on correspondingly changing populations. Like **RQ2**, **RQ3** generalizes **RQ1** in the sense that **RQ3** is equivalent to **RQ1** when $\tilde{T} = 1$. We can assume $T = 1$ for **RQ3** (a), since there is no point in retraining on the sample because there are no performative effects on the sample, while $T$ can be any positive integer in **RQ3** (b). Unlike **RQ2**, however, **RQ3** is only relevant if the whole population $d_t$ is available to the institution.

(**RQ4**) The last research question is *statistical* in nature. It compares RERM $\{\widehat{\theta}_t\}_{t=1}^T$ to a simultaneous (*hypothetical*) RRM $\{\theta_t\}_{t=1}^T$ on the population. Can we bound the *inferential gap* $\sum_{t=1}^T (\mathscr{R}(d_t, \widehat{\theta}_t) - \mathscr{R}(d_t, \theta_t))$ between the two? This question is partly answered by the central limit theorem (CLT) on statistical inference under *stateless* performativity by Li et al. (2025b), which states that, for any $t$, $\sqrt{n}(\widehat{\theta}_t - \theta_t) \xrightarrow{D} N(0, \Sigma_t)$ as $n \to \infty$, where $\Sigma_t$ is an identifiable covariance matrix. **RQ4** gives the standard CLT (de Moivre, 1738) for $t = 1$ as a special case. Obviously, this result implies that the inferential gap goes to zero as $n \to \infty$, under continuity of the risk. Finite sample analysis and the extension to the *stateful* case are yet to be conducted.

## 3. Generalization Bounds on Performative Predictions

The previous section provides a conceptual understanding of what generalization means in a performative world. We now turn to a technical embedding of performativity into statistical learning theory (Vapnik & Chervonenkis, 1968; Vapnik, 1991; 1998; 1999). We answer **RQ1**–**RQ3** under a generic assumption on the nature of performative shifts, namely Wasserstein sensitivity of the transition map Tr (Condition 3.2). This assumption has been used consistently throughout the literature[1] and we refer to Kim & Perdomo (2022) for a discussion of its generality as well as to Section 4 for an illustration of this assumption in the context of running example (B).[2] Specifically, we do not assume any

particular functional form of Tr like, e.g., Mendler-Dünner et al. (2022); Kirev et al. (2025). The price we pay for this generality is the fact that our bounds are looser than they would be otherwise (yet still insightful, see Corollary 3.11). To address this, we will include more conditions to obtain tighter bounds later. To begin with, however, we only need the following subset of conditions in Perdomo et al. (2020); Brown et al. (2022):[3]

**Condition 3.1.** The loss $\ell(z, \theta)$ is $\gamma$-strongly convex in $\theta$.

**Condition 3.2.** The transition map Tr is $(\varepsilon, p)$-jointly sensitive for some $\varepsilon > 0$ and some $1 \leq p \leq \max\{2, \nu/2\}$. That is, $W_p(\text{Tr}(d, \theta), \text{Tr}(d', \theta')) \leq \varepsilon W_p(d, d') + \varepsilon \|\theta - \theta'\|_2$, where $W_p$ denotes the $p$-Wasserstein distance.

**Condition 3.3.** The loss $\ell(z, \theta)$ is $L_\ell$-Lipschitz in $z$ and $\kappa$-continuously differentiable with respect to $\theta$—i.e., at every $z \in \mathcal{Z}$, the gradient $\nabla_\theta \ell(z, \theta)$ exists and is $\kappa$-Lipschitz continuous (with respect to the $L_2$ norm on domain and codomain) in each of $\theta$ and $z$.

Condition 3.3 implies that the function $G$ (Definition 2.1) is $L_a$-Lipschitz with $L_a > 0$ on the convex and compact $\Theta$ (see Appendix D.1 and Lemma 2 of Brown et al., 2022). Our strategy for proving generalization bounds is twofold. In order to generalize from $\widehat{d}_T$ to $d_0$ (**RQ1**), we bound the Wasserstein distances $W_p(\widehat{d}_0, d_0)$ (Lemma 3.4) and $W_p(\widehat{d}_0, \widehat{d}_T)$ (Lemma 3.5), before relating these divergence bounds to expectation difference bounds via the Kantorovich-Rubinstein Lemma (Kantorovich & Rubinstein, 1958).

**Lemma 3.4** (Convergence in Wasserstein Space; Fournier & Guillin 2015). *Let $\mathscr{D}_{\mathcal{Z}} := \sup_{z,z'} \|z - z'\|_2 < \infty$. Then, for any $\beta_0 \in (0, \infty)$, we have*

$$W_p\left(\widehat{d}_0, d_0\right) \leq \beta_0,$$

*with probability of at least $1 - C_a \exp\left(-C_b n \beta_0^\nu\right)$, where $C_a$ and $C_b$ are constants that depend on $p, \nu$ and $\mathscr{D}_{\mathcal{Z}}$ only.*

**Lemma 3.5** (In-Sample Performative Shift Bound). *Assume that at most $m$ units (in the sample of size $n$) change in response to predictions at each iteration $t$.[4] If Conditions 3.1–3.3 hold, we have*

$$W_p(\widehat{d}_0, \widehat{d}_T) \leq \frac{[\varepsilon(1 + L_a)]^T - 1}{\varepsilon(1 + L_a) - 1} \left(\frac{m}{n}\right)^{\frac{1}{p}} \mathscr{D}_{\mathcal{Z}},$$

---

[1]See Definition 3.1 in Perdomo et al. (2020), Equation (A1) in Miller et al. (2021), Definition 1 in Brown et al. (2022), the even more restrictive (A1) in Mofakhami et al. (2023), Assumption 3.1(d) in Li et al. (2025b), Definition 1 in Mendler-Dünner et al. (2020), Assumption 1.2 in Jagadeesan et al. (2022), Assumption A3 in Li et al. (2022), Assumption 2 in Narang et al. (2023), Assumption 2.2 in Wang et al. (2023), Condition W1 in Li & Wai (2024), Definition 2.3 in Jin et al. (2026),

[2]In the running example (A) of routing apps, Wasserstein sensitivity would, e.g., mean that the next-round traffic behavior is

generated by a Markov kernel that is Lipschitz in both $z$ and $\theta$. This is one type of response studied in routing settings where recommendation changes induce bounded changes in traffic behavior. (Cabannes, 2019; Benenati & Grammatico, 2024)

[3]While Conditions 3.1 and 3.2 are used throughout Perdomo et al. (2020); Brown et al. (2022), Condition 3.3 is only required for Theorem 4.3 and Corollary 5.1 in Perdomo et al. (2020) and Theorem 5 and 6 in Brown et al. (2022).

[4]Formally, denote by $m_t = n \sup_A \left|\widehat{d}_t(A) - \widehat{d}_{t-1}(A)\right|$ the number of units changing at iteration $t \geq 1$ (where the supremum is over all events $A$). Then, $m = \max\{m_1, \ldots, m_T\}$.

*pointwise in $\widehat{d}_0$ (i.e., for any fixed $\widehat{d}_0$).*[5]

Lemma 3.4 is an application of Theorem 2, Case 1 in Fournier & Guillin (2015) with $\mathcal{E}_{\alpha,\gamma}(\mu) < \infty$ and $\alpha = \nu$. It implies $\widehat{d}_0$ is arbitrarily close to $d_0$ in Wasserstein space as $n \to \infty$ with high probability. Proofs of all our results, including Lemma 3.5, can be found in Appendix D.

**RQ2–RQ4** involve performative shifts of the true law $d_0$, which implies we might have to evaluate $f_\theta$ outside the support of $d_0$. This is why we choose covering numbers (Kolmogorov & Tikhomirov, 1959; Talagrand, 2021) over popular Rademacher or Gaussian complexities (Bartlett & Mendelson, 2002) as a complexity measure to quantify the richness of the hypothesis class $\mathcal{F}$.

**Definition 3.6** (Covering Number Entropy Integrals). Let $\|\cdot\|$ denote a norm on $\mathcal{F}$, such as the uniform norm

$$\|f\|_{L_\infty(d)} = d\text{-ess} \sup_{x \in X} \|f(x)\|_2,$$

or the $L_2$ norm

$$\|f\|_{L_2(d)} = \sqrt{\mathbb{E}_{X \sim d}[\|f(X)\|_2^2]}$$

with respect to $d \in \Delta$. For $\epsilon > 0$, define the covering number $\mathcal{N}(\mathcal{F}, \|\cdot\|, \epsilon)$ as the minimal $N$ such that there exists $\theta_1, \ldots, \theta_N \in \Theta$ satisfying $\min_{1 \le k \le N} \|f_\theta - f_{\theta_k}\| \le \epsilon$ for all $\theta \in \Theta$. The covering number entropy integrals for the uniform and $L_2$ norm are

$$\mathfrak{C}_{\infty(d)}(\mathcal{F}) := \int_0^\infty \sqrt{\log \mathcal{N}\left(\mathcal{F}, \|\cdot\|_{L_\infty(d)}, \varepsilon\right)} \mathrm{d}\varepsilon,$$

$$\mathfrak{C}_{L_2}(\mathcal{F}) := \sup_{d \in \Delta} \int_0^\infty \sqrt{\log \mathcal{N}\left(\mathcal{F}, \|\cdot\|_{L_2(d)}, \varepsilon\right)} \mathrm{d}\varepsilon.$$

### 3.1. RQ1: Generalization Under Sample Performativity

We are now ready to answer **RQ1** with the following result.

**Theorem 3.7** (Excess Risk Bound, **RQ1**). *The excess risk $\mathcal{R}(d_0, \widehat{\theta}_T) - \inf_{\theta \in \Theta} \mathcal{R}(d_0, \theta)$ of a model $\widehat{\theta}_T$ (re)trained on performative samples $\widehat{d}_0, \ldots, \widehat{d}_{T-1}$, in which at most $m \le n$ units change in response to predictions, is upper bounded by*

$$L_\ell \left(\frac{\log(C_a/\delta)}{C_b n}\right)^{\frac{1}{\nu}} + L_\ell L_a \frac{[\varepsilon(1+L_a)]^{T-1} - 1}{\varepsilon(1+L_a) - 1} \left(\frac{m}{n}\right)^{\frac{1}{p}} \mathscr{D}_{\mathcal{Z}}$$
$$+ \frac{2FL_\ell}{\sqrt{n}} \left(12\mathfrak{C}_{L_2}(\mathcal{F}) + \sqrt{2\ln(1/\delta)}\right),$$

*for any $T \in \mathbb{N}$, with probability over $d_0$ of at least $1 - 2\delta$, under Conditions 3.1–3.3.*

---

[5]With the usual convention for the geometric fraction $\frac{[\varepsilon(1+L_a)]^T - 1}{\varepsilon(1+L_a) - 1} = T$ if $\varepsilon(1+L_a) = 1$.

Our setting's generality requires a generic proof strategy: We bound $\mathcal{R}(d, \widehat{\theta}_T) - \mathcal{R}(\widehat{d}_0, \widehat{\theta}_T)$, $\mathcal{R}(\widehat{d}_0, \widehat{\theta}_T) - \mathcal{R}(\widehat{d}_0, \widehat{\theta}_0)$ and $\mathcal{R}(\widehat{d}_0, \widehat{\theta}_0) - \inf_{\theta \in \Theta} \mathcal{R}(d, \theta)$ with high probability over $d_0$ and then combine via the union bound (see Appendix D.2 for details and the complete proof).

We can already make three interesting observations. First, for fixed $T$, the bound in Theorem 3.7 goes to zero as $n \to \infty$ if and only if $m = o(n)$. Second, the size of the second term ("performative term") is governed by $\varepsilon(1 + L_a)$: it grows exponentially in the number of performative sample updates $T$ if $\varepsilon(1 + L_a) > 1$, linearly in $T$ if $\varepsilon(1 + L_a) = 1$, and is bounded otherwise. Thus, $\varepsilon(1 + L_a)$ controls how much repeated sample performativity amplifies the finite-sample bound. Third, the bound generally grows in $m$, hinting at a fundamental trade-off between manipulating a sample and learning from it. For instance, if the job center in Example (B) wants to help more unemployed people (i.e., increase $m$) among their clients $n$, this comes at the cost of generalizing their model to new, unseen clients. Section 4 will dive further into this trade-off. Intuitively, this trade-off also holds beyond the setting of Theorem 3.7, because changing more units of an i.i.d. sample in an unknown way cannot guarantee improved generalization. Besides, our excess risk bound directly implies a data-dependent bound on the generalization error (see proof in Appendix D.3):

**Corollary 3.8.** *In the setting of Theorem 3.7 (i.e., under Conditions 3.1–3.3) $\mathcal{R}(d_0, \widehat{\theta}_T) - \mathcal{R}(\widehat{d}_{T-1}, \widehat{\theta}_T)$ is upper bounded by*

$$L_\ell \left(\left(\frac{\log(C_a/\delta)}{C_b n}\right)^{\frac{1}{\nu}} + \frac{\varepsilon^T (1+L_a)^{T-1} - 1}{\varepsilon(1+L_a) - 1} \left(\frac{m}{n}\right)^{\frac{1}{p}} \mathscr{D}_{\mathcal{Z}}\right)$$

*with probability over $d_0$ of at least $1 - 2\delta$.*

### 3.2. RQ2: Generalization Under Full Performativity

In order to answer **RQ2**, we bound the excess risk of learning under full (i.e, sample and population) performativity. Truth is elusive now: The learning target is no longer static, but changes from $d_0$ to $d_1, d_2, \ldots$ in response to predictions. As a first step, we thus have to bound $W_p(d_0, d_T)$ in addition to $W_p(\widehat{d}_0, d_0)$ (Lemma 3.4) and $W_p(\widehat{d}_0, \widehat{d}_T)$ (Lemma 3.5).

**Lemma 3.9** (Performative Population Shift Bound). *Assume $s \in [0, 1]$ is the share of units in $d_0$ reacting to predictions (the "performative response rate"). Then*

$$s < \frac{m}{n} + q(\delta)\sqrt{\frac{m}{n^2}\left(1 - \frac{m}{n}\right)},$$

*with probability $1 - \delta$, where $q(\delta)$ is the $(1-\delta)$-quantile of the standard normal distribution.*

This lemma follows directly from Wald's method (Brown et al., 2001) by treating the observed fraction $m/n$ as an

estimator of an unobserved Bernoulli parameter $s$. We can now bound the excess risk under full performativity.

**Theorem 3.10** (Performative Excess Risk Bound, **RQ2**). *Under Conditions 3.1–3.3, the performative excess risk $\mathscr{R}(\mathrm{Tr}(d_0, \widehat{\theta}_T), \widehat{\theta}_T) - \inf_{\theta \in \Theta} \mathscr{R}(\mathrm{Tr}(d_0, \widehat{\theta}_T), \theta)$ of $\widehat{\theta}_T$ is upper bounded by $A(m,n) + L_\ell \Big( C(n) + \frac{2F}{\sqrt{n}} \Big( 12\mathfrak{C}_{L_2}(\mathcal{F}) + \sqrt{2\ln(1/\delta)} \Big) + L_a K(T,m,n) \Big)$, with probability greater than $1 - 4\delta$. The terms $A(m,n)$, $C(n)$ and $K(T,m,n)$ depend only on constants and $m$, $n$ and $T$.*

The proof in Appendix D.5 provides explicit expressions for $A(m,n)$, $C(n)$ and $K(T,m,n)$, which show that the bounds in Theorems 3.7 and 3.10 grow at similar rates. Crucially, these theorems point to two ways of improving generalization (guarantees) under performativity. The first one is straight-forward: I) Without knowledge of $\mathrm{Tr}$, retraining on $\widehat{d}_0, \dots, \widehat{d}_T$ results in models that generalize worse than initial $\widehat{\theta}_0$, because the excess risk bounds grow in $T$, both via $m = \max\{m_1, \dots, m_T\}$ (see footnote to Lemma 3.5) and via the geometric factor $\frac{[\varepsilon(1+L_a)]^T - 1}{\varepsilon(1+L_a)-1}$, which grows exponentially in $T$ when $\varepsilon(1 + L_a) > 1$. The second one is less intuitive: II) Retraining on $\widehat{d}_0, \dots, \widehat{d}_T$ can still improve generalization bounds. While it worsens $\widehat{\theta}_t$'s generalization capabilities, it allows for estimating $\mathrm{Tr}$ from observed $\widehat{d}_1, \dots, \widehat{d}_T$ more efficiently than Lemma 3.9. This tightens the bound in Theorem 3.10. Lemma 3.9 conservatively relies on $m$ from that iteration, in which the most units react. But we have more information, as we observe the number $m_t$ of units changing at each $t$ (see footnote to Lemma 3.5).

**Corollary 3.11** (Improving Bounds Under Performativity). *I) $\widehat{\theta}_0$ yields the tightest performative excess risk bound among $\{\widehat{\theta}_t\}_{t=0}^T$.
II) It holds with probability $1 - \delta$ that*

$$s < \frac{M_t}{nT} + q(\delta)\sqrt{\frac{M_T}{n^2 T^2}\left(1 - \frac{M_T}{nT}\right)},$$

*where $M_T = \sum_{t=1}^T m_t$, leading to an as-tight or tighter performative excess risk bound as the one in Theorem 3.10.*

Corollary 3.11 constitutes a remarkably simple and readily applicable insight: If predictions exhibit performative effects on both the sample and the population, use I) the initial fit $\widehat{\theta}_0$ on out-of-sample data and II) estimate the performative shift it will cause from the sample shifts you have observed, in order to obtain the tightest generalization guarantees.

These bounds, however, will still exhibit some slack due to their generality. This naturally begs the question whether we can prove tighter bounds under slightly stricter assumptions than Conditions 3.1–3.3. The following two results give affirmative answers. While Theorem 3.13 requires the strong Condition 3.12 of all functions in $\mathcal{F}$ to be Lipschitz, Theorem 3.15 makes do with the substantially weaker Condition 3.14. Both Theorems bound the generalization gap, but a bound on the respective excess risk directly follows from the proofs of Theorems 3.7 and 3.10 (see Appendix D).

**Condition 3.12** (Regularity of Models). *All functions $f_\theta$ in $\mathcal{F}$ are upper semi-continuous and $L_f$-Lipschitz.*

**Theorem 3.13** (Generalization Gap I, **RQ2**). *Under Conditions 3.1–3.3 and 3.12, the performative generalization gap $\mathscr{R}(\mathrm{Tr}(d_0, \widehat{\theta}_T), \widehat{\theta}_T) - \mathscr{R}(\mathrm{Tr}(\widehat{d}_0, \widehat{\theta}_T), \widehat{\theta}_T)$ is upper bounded with probability $1 - 3\delta$ over $d_0$ by*

$$\frac{48}{\sqrt{n}}\left(\mathfrak{C}_{\infty(d_0)}(\mathcal{F}) + \frac{L_\ell L_f \mathscr{D}_\mathcal{Z}{}^p}{R^{p-1}}\right) + F\sqrt{\frac{2\log(2/\delta)}{n}} + 2L_\ell R,$$

*where $R$ is the maximum of the bounds in Lemmas 3.5 and 3.9.*

**Condition 3.14** (Weaker Regularity). *There exist $f_0 \in \mathcal{F}$, $B \geq 0$, $\mathfrak{d} \in \mathbb{N}$ and $x_0 \in \mathcal{X}$, such that $\ell(f_0(x), y) \leq B\|x - x_0\|_2^{\mathfrak{d}}$ for all $x \in \mathcal{X}$ and $y \in \mathcal{Y}$.*

**Theorem 3.15** (Generalization Gap II, **RQ2**). *Under Conditions 3.1–3.3 and 3.14, we obtain the same bound on $\mathscr{R}(\mathrm{Tr}(d_0, \widehat{\theta}_T), \widehat{\theta}_T) - \mathscr{R}(\mathrm{Tr}(\widehat{d}_0, \widehat{\theta}_T), \widehat{\theta}_T)$ as in Theorem 3.13 but with $B2^{p-1}(1 + \mathscr{D}_\mathcal{Z}/R)^p$ instead of $L_\ell L_f R^{1-p}$.*

The key technique in both proofs is to leverage empirical process theory for dual characterizations of locally inf-sup and inf-inf risk functionals in Wasserstein space (Lee & Raginsky, 2018; Gao & Kleywegt, 2023). Such functionals are commonly used in distributionally robust optimization (DRO) (Shapiro et al., 2021) and distributionally favorable optimization (Jiang & Xie, 2025), respectively. In PP, they arise from the fact that we do not assume the transition map $\mathrm{Tr}$ to have a particular functional form, but we can nevertheless bound the shift induced by $\mathrm{Tr}$. The worst-case generalization error of $\widehat{\theta}_T$ then is $\mathscr{R}(\mathrm{Tr}(d_0, \widehat{\theta}_T), \widehat{\theta}_T) = \sup_{d \in \mathcal{A}} \mathscr{R}(d, \widehat{\theta}_T)$, where $\mathcal{A} = \{d : W_p(d_0, d) \leq b\}$ with $b$ a bound on the shift induced by $\mathrm{Tr}$ implied by Lemma 3.9. The conceptual difference to DRO is that the worst case generalization *gap* between $\mathscr{R}(\mathrm{Tr}(d_0, \widehat{\theta}_T), \widehat{\theta}_T)$ and $\mathscr{R}(\mathrm{Tr}(\widehat{d}_T, \widehat{\theta}_T), \widehat{\theta}_T)$ under performativity does not only entail the worst case (supremum) risk on the population, but also the best case (infimum) risk on the sample (see Equation 24 in the proof of Theorem 3.13).

**Structural insight:** In this worst case, the sample "deceives" you by producing *self-fulfilling* performative effects, while the population acts in a *self-negating* way. In other words, RERM corresponds to fitting self-fulfilling models[6] here, creating an empirical echo chamber, which results in models

---

[6]That is, solving $\arg\inf_\theta \inf_{d \in \mathcal{A}'} \mathscr{R}(d, \theta)$ with $\mathcal{A}' = \{d : W_p(\widehat{d}_0, d) \leq b' \wedge |\mathrm{supp}(d)| < \infty\}$ and $b'$ informed by Lemma 3.5.

"far away" from good ones on the population, since the latter reacts in the opposite way. In Example (A) of routing apps, this means that drivers in San Francisco (the sample) fully trust the app and change their behavior to fulfill the predictions, while Bay Area drivers (the population) do the exact opposite and negate the predictions.

### 3.3. Cumulative Performative Excess Risk Bound, RQ3

In addition to the error when generalizing from the sample to the performatively shifted population as in **RQ2** above, **RQ3** asks for the error when we retrain the model on this shifted population. In this scenario, the institution first trains a model $T$ times on a sample before rolling it out to the population and retraining there $\tilde{T} - T$ times. Crucially, the performative effect of the (sample-trained) model $\widehat{\theta}_T$ on the population happens *before* the the model is trained on the population for the first time, see Section 2. For details on (and the proof of) Theorem 3.16, we refer to Appendix D.9.

**Theorem 3.16** (Cumulative Performative Excess Risk Bound). *Under Conditions 3.1–3.3, we have that,*

$$
\sum_{t=T}^{\tilde{T}} \Big( \mathscr{R}(d_t, \theta_t) - \inf_{\theta \in \Theta} \mathscr{R}(d_t, \theta) \Big) \leq \mathscr{B}(T, m, n) +
$$

$$
(\tilde{T} - T + 1) \, L_\ell L_a \left( \frac{m}{n} + q(\delta) \sqrt{\frac{\frac{m}{n}(1 - \frac{m}{n})}{n}} \right)^{\frac{1}{p}} \mathscr{D}_{\mathcal{Z}},
$$

*for any $\tilde{T} \geq T$, with probability $1 - 5\delta$, where $d_t = \mathrm{Tr}(d_{t-1}, \theta_t)$ for $t \geq T$, $\theta_T = \widehat{\theta}_T$, $d_{T-1} = d_0$ and $\mathscr{B}(T, m, n)$ is the performative excess risk bound from Theorem 3.10.*

**Key takeaways:** Our analysis identifies three consequences of learning under performativity. First, the share of sampled units whose data change after predictions are deployed directly controls the finite-sample excess-risk and performative excess-risk bounds (Theorems 3.7 and 3.10). Second, learning under performativity exhibits a change–learn trade-off: the more a model changes the data from which it learns, the harder it becomes to generalize from the resulting sample. Third, in the worst case, the sample can self-fulfill predictions while the population self-negates them, creating an empirical echo chamber that inflates the performative generalization gap (Theorems 3.13 and 3.15). Corollary 3.11 shows that performatively shifted samples are nevertheless useful, since repeated deployments can help estimate population-level performative shifts.

## 4. Illustration of Bounds on Job Seeker Data

To demonstrate how our generalization bounds look in practice, we illustrate them on administrative data from the German Federal Employment Agency. The dataset con-

tains a 2% sample of German labor market records spanning 1975 to 2017, containing over 60 million rows in its raw form. We emphasize that the goal here is a mere *illustration* rather than an application of our bounds. The latter would require historical records of predictions that are usually unavailable (Mendler-Dünner et al., 2022). We consider binary prediction tasks: will a job seeker remain continuously unemployed for some period of time? Such predictions typically have performative effects on whether people actually remain unemployed, since only those with high predicted risks of staying unemployed receive job trainings that help them to actually find a job (see Junquera & Kern, 2025, and Section 1.2). The administrative data set contains 28 features on labor market histories, education, skill level and demographic information. To illustrate our bounds, we use a Lipschitz continuous (Condition 3.12) logistic regression $f_\theta^{\log}(x) = \sigma(\theta^\top x)$, where $\sigma(t) := (1 + e^{-t})^{-1}$ with continuously differentiable (Condition 3.3) logistic loss $-\big(y \log f_\theta(x) + (1 - y) \log(1 - p_\theta(x))\big)$, to assign job trainings, triggering jointly sensitive (Condition 3.2) performative shifts. We employ standard $L_2$-regularization, which renders the loss strongly convex (Condition 3.1). All details on data pre-processing, model training and bound computation can be found in Appendix C.[7]

**Historical Data (Sample Performativity, RQ1):** We exemplarily choose a sampled cohort of $n = 60147$ German residents $\widehat{d}_0 \overset{\text{i.i.d.}}{\sim} d_0$, where $t = 0$ corresponds to losing their job in 2012. Their job center assigns some of them to trainings based on a fitted $f_{\widehat{\theta}_1}^{\log}$, performatively shifting $\widehat{d}_0$ to $\widehat{d}_1 = \mathrm{Tr}(\widehat{d}_0, \widehat{\theta}_1)$, where $t = 1$ corresponds to 14 days after losing their job, where we observe that $m = 1816$ residents change at least one feature. Refitting on this $\widehat{d}_1$ gives $f_{\widehat{\theta}_2}^{\log}$, whose generalization gap $\mathscr{R}(d_0, \widehat{\theta}_2) - \mathscr{R}(\widehat{d}_1, \widehat{\theta}_2)$ we can upper bound by $\approx 0.01 + 0.29 = 0.3$ with probability 0.95 by Corollary 3.8; the first summand is the sampling term (small due to large $n$), the second is driven by the observed performative response rate $m/n = 1816/60147$. The bound tells the job center that the generalization error of their model will not exceed the training error by more than 0.3 nats (logistic loss units) with 95% confidence.

**Semi-Simulation (Full Performativity, RQ2):** On another sampled cohort $\widehat{d}_0 \overset{\text{i.i.d.}}{\sim} d_0$ of $n = 41585$ German residents that just lost their jobs in 2012, we set $t = 1$ to 60 days after losing their job. We simulate assignments to job trainings based on predicted risks of prolonged unemployment by fitted logistic regression $f_{\widehat{\theta}_1}^{\log}$. We assign the $\xi n$ units with the highest predicted risks among the initial sample to job trainings with policies $\xi \in \{0.01, 0.02, \ldots, 0.5\}$. For ease of exposition, we assume these trainings to be fully effec-

---

[7]Code to replicate our case study is available at https://github.com/rodemann/plt-jobseekers.

tive[8], i.e., flip targets from `jobless` to `employed` for all $\xi n$ units, and no further changes, i.e., $m = \xi n$. Retraining on so shifted samples $\widehat{d}_{1,\xi}$ gives $f_{\widehat{\theta}_{2,\xi}}^{\log}$. We then simulate the employment agency to roll out the risk-based assignments to all of Germany $d_0$, again with varying $\xi$, triggering a population shift $d_{1,\xi} = \text{Tr}_\xi(d_0, \widehat{\theta}_2)$. Theorem 3.13 allows bounding the performative generalization gap $\mathscr{R}(d_{1,\xi}, \widehat{\theta}_{2,\xi}) - \mathscr{R}(\text{Tr}(\widehat{d}_1, \widehat{\theta}_{2,\xi}), \widehat{\theta}_{2,\xi})$ as a function of the agency's policy $\xi$. Figure 1.3 shows the total bound that holds with probability of $1 - \delta = 0.95$ for varying $\xi$, along with the bound's components $\frac{48}{\sqrt{n}} \left( \mathfrak{C}_{\infty(d_0)}(\mathcal{F}) L_\ell L_f R^{1-p} \mathscr{D}_{\mathcal{Z}}^p \right)$ (adaptive complexity term), $2 L_\ell R$ (performative term) and $F \sqrt{(2 \log(2/\delta))/n}$ (sampling term). The more people receive job trainings, the less reliable the model's predictions become due to performative shifts both in and out of the sample—an instantiation of the principled trade-off between learning and changing we found in Section 3.

## 5. Related Work, Limitations and Outlook

Table 1 simultaneously facilitates positioning our work within the body of related literature, as well as an outlook to future work. For example, Table 1 hints at Li et al. (2025b) answering **RQ4** for $n \to \infty$, as detailed in Section 2. Thus, answering **RQ4** for finite $n$ is a promising avenue for future work.

**Related work:** Besides those works already discussed in Section 2, *calibration* under performativity (Li et al., 2025a; Boeken et al., 2025; Zalouk et al., 2025) is conceptually related to *generalization* under performativity, see Appendix B. On a technical level, our analysis is related to distributionally robust performative optimization (Jia et al., 2025) and prediction (Xue & Sun, 2024). These works use Wasserstein ambiguity sets to robustify optimization and prediction, respectively, against performative distribution shifts. Besides studying generalization instead of prediction or optimization, our work differs in the way ambiguity sets are specified: we estimate them from the sample's performative reactions instead of specifying them *a priori*. Appendix B has more in-depth discussion of (further) related work.

**Limitations and General Discussion:** This article revealed an explicit trade-off between changing the world and learning from it: The more a model affects data, the less is to be learned from it. We further showed that so-changed samples can still be useful in estimating the performative shifts out of sample under natural assumptions, giving rise to a *practical take-away:* If you retrain under performativity, combine the initial model and the information from all performatively shifted samples to get the tightest generalization bound, i.e., the best guarantee for out-of-sample application of your

model under performativity. On a more principled level, our bounds show that generalizing in a performative world is hard for two reasons: Not only can the population invalidate your predictions (*self-negation*), the sample might also deceive you by doing the exact opposite and confirm your predictions (*self-fulfilling*), effectively creating an empirical echo chamber. Our bounds require Conditions 3.1–3.3 from the original PP framework (Perdomo et al., 2020; Brown et al., 2022). Condition 3.1 (strongly convex loss) is arguably restrictive. For tighter bounds, we even need that $\mathcal{F}$ contains at least one $f_0$ that is Lipschitz. However, we emphasize that these assumption refer to something (loss and model class) we have control over. In other words, they are *verifiable*. Their strength allows us to abstain from specific assumptions on unknown $\text{Tr}$, which we have no control over and whose properties we can only estimate.

**Outlook:** This trade-off (restricting the models the analysis applies to vs. making assumptions about the unknown performative shift) leaves room for future work on the other side of this trade-off. For example, adopting the setup in Mofakhami et al. (2023) allows us to drop the convexity condition on $\ell$ at the price of stronger assumptions on $\text{Tr}$.

## 6. Summary and Conclusion

We initiated the study of generalization under performativity, where predictive models must generalize from samples to populations that may themselves react to those predictions. While classical performative prediction focuses on stability and optimality under population-level feedback, our focus was (finite-sample) learning under performativity: whether models trained on samples can still generalize when the sample, the population, or both react to deployed predictions. This distinction led to separate notions of sample performativity, population performativity, and full performativity. We embedded performative predictions into statistical learning theory and gave generalization bounds on performative predictions. Specifically, we derived bounds on excess risk, performative excess risk, generalization gaps, and cumulative performative excess risk.

At a high level, our results show that learning under performativity is governed by a trade-off between intervention and inference: the more predictions change the data, the harder it becomes to generalize from the resulting data. In the worst case, the sample self-fulfills predictions while the population self-negates them, creating an empirical echo chamber. At the same time, performatively shifted samples remain useful because they help estimate out-of-sample performative shifts. All in all, *performative learning theory* provides a first set of tools for analyzing when predictive systems can still learn from data they themselves help create.

---

[8]Otherwise, our bounds would still be valid, since $\xi n$ is an upper bound for the number of changed units either way.

## Acknowledgements

The authors thank Juan C. Perdomo (MIT), Rabanus Derr (University of Tübingen), Jonas Hanselle (LMU Munich), Huynh Quang Kiet Vo (CISPA), Gowtham Reddy Abbavaram (CISPA) and Anurag Singh (CISPA) for helpful comments, stimulating discussions as well as valuable feedback at various stages of this work. We also thank the four anonymous reviewers for their assessment of our paper and their helpful feedback. JR gratefully acknowledges support by the LMU Mentoring Program within the Faculty of Mathematics, Statistics and Computer Science at LMU Munich and the Bavarian Academy of Sciences (BAS). JR further acknowledges funding by the federal German Academic Exchange Service (DAAD) supporting a research stay at Harvard University. UFA acknowledges the support by the DAAD programme Konrad Zuse Schools of Excellence in Artificial Intelligence, sponsored by the Federal Ministry of Education. This work was partially conducted while JB was a PhD student in the Department of Statistics at Harvard University. JB gratefully acknowledges partial financial support from the Australian-American Fulbright Commission and the Kinghorn Foundation during his time as a PhD student.

## Impact Statement

Performative predictions (by definition) have societal consequences. The overall research direction of performativity aims to understand and mitigate potential harms that arise when machine learning systems shape the very outcomes they predict. By developing theoretical foundations for generalization under performativity, our line of work contributes to more responsible deployment of predictive systems in high-stakes domains such as employment services, education, criminal justice, and finance.

Our work reveals a trade-off between performatively changing the world and learning from it, which has important implications for practitioners: systems that heavily influence their environment may simultaneously undermine their own ability to make accurate predictions about that environment. This insight could inform more cautious deployment strategies and highlight the need for careful monitoring when predictive systems are rolled out.

The specific generalization bounds we derive could help institutions better understand the limitations of their models under performative effects, potentially preventing overconfident deployment decisions that could harm vulnerable populations. For instance, our analysis of job training allocation (Section 4) demonstrates how performative effects complicate generalization in unemployment prediction systems. The latter constitutes a domain where prediction errors can have significant consequences for individuals' livelihoods.

Beyond this illustrative use case in Section 4, however, the paper at hand makes fundamental theoretical contributions and is very unlikely to have *direct* ethical or social consequences. The work is primarily mathematical in nature and does not introduce new applications. Any societal impact would be indirect, i.e., mediated through how practitioners and researchers use these theoretical insights to design and deploy performative prediction systems.

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

## A. Generalization Bounds on Stable and Optimal Models

Under appropriate regularity conditions (strong convexity and smoothness of the loss function, as well as Lipschitz continuity of the distribution mapping), the sequence of RRM converges to a *stable pair* $(\theta_S, d_S)$ where $d_S = \mathrm{Tr}(d_S, \theta_S)$ is a fixed-point distribution for $\theta_S$ and $\theta_S \in G(d_S)$. Brown et al. (2022, Theorem 6) further show that $\theta_S$ is close to the optimal $\theta_{\mathrm{OPT}}$, which is defined as the parameter $\theta$ that achieves the minimal long-run loss $\mathbb{E}_{Z \sim d_\theta} \ell(Z, \theta)$ on its fixed-point distribution $d_\theta$ resulting from repeatedly applying $\mathrm{Tr}$ to $\theta$ ($d_\theta$ is unique—i.e., it does not depend on the starting distribution $d_0$—due to the Banach fixed-point theorem). In other words, R(E)RM under performativity might retrieve stable models $\widehat{\theta}_S$ (stable on sample) and $\theta_S$ (stable on population) or even optimal ones $\widehat{\theta}_{\mathrm{OPT}}$ and $\theta_{\mathrm{OPT}}$ under conditions detailed in Perdomo et al. (2020); Brown et al. (2022).

These models are subsumed by our analysis. They constitute special cases of $\widehat{\theta}_t$ and $\theta_t$, respectively. For instance, the performative excess risks of stable and optimal models $\widehat{\theta}_S$ and $\widehat{\theta}_{\mathrm{OPT}}$ are

$$\mathscr{R}(\mathrm{Tr}(d_0, \widehat{\theta}_S), \widehat{\theta}_S) - \inf_{\theta \in \Theta} \mathscr{R}(\mathrm{Tr}(d_0, \widehat{\theta}_S), \theta),$$

and

$$\mathscr{R}(\mathrm{Tr}(d_0, \widehat{\theta}_{\mathrm{OPT}}), \widehat{\theta}_{\mathrm{OPT}}) - \inf_{\theta \in \Theta} \mathscr{R}(\mathrm{Tr}(d_0, \widehat{\theta}_{\mathrm{OPT}}), \theta),$$

respectively.

## B. Further Related Work

Recent years have seen several articles tangential to generalization under performativity. As mentioned above, Li et al. (2025a); Boeken et al. (2025); Zalouk et al. (2025) study calibration under performativity: They investigate the discrepancy between predicted and true *conditional probabilities* under performativity, whereas our work studies the discrepancy between empirical and true *risk* under performativity.

Zhang & Conitzer (2021) study generalization under strategic reactions to predictions, i.e., performative shifts of $X$. Kirev et al. (2025) study PAC learnability of binary classification under (linear) performative shifts of $Y \mid X$ and marginal $X$, which is a special case of **RQ2** (a) (see Table 1). Our framework is more general than both these works in two ways. First, it accounts for all Lipschitz continuous transition maps, as in Perdomo et al. (2020). Second, it studies performative shifts on any subsets of $Y$ and $X$.

Mendler-Dünner et al. (2022) study the causal effects of predictions on outcomes $Y$. If valid causal estimates can be obtained from (typically) observational data, they can improve generalization (see also König et al., 2025). Akin to the trade-off between performatively changing a sample and learning from it, Wilder & Welle (2025) identify a trade-off between treating those in need and learning about population level quantities. Generally, causal inference (Pearl, 2009; Peters et al., 2017) can model performative effects through structural causal models, although it typically requires strong identifiability assumptions. Our approach treats $\mathrm{Tr}$ as a black box, connecting to work on distribution shift (Quiñonero Candela et al., 2009; Shimodaira, 2000; Gama et al., 2014) and domain adaptation (Muandet et al., 2013; Zhang et al., 2013). However, unlike passive distribution shift, performative prediction involves active shift where the learner's model causes the change. Recent advances in learning under distribution shift via kernel mean embeddings (Muandet et al., 2017; 2021) and domain generalization (Muandet et al., 2013; Blanchard et al., 2021) share our distributional perspective, but our focus is on finite-sample bounds under performative feedback rather than adapting across fixed domains.

Technically related, Rodemann et al. (2024) and Rodemann & Bailie (2025) study learning from samples that are adaptively shifted by models (see also Hazan et al., 2025). While relying on Wasserstein ambiguity sets just like in Theorems 3.13 and 3.15, their work is conceptually different: the models there have full control over samples, while performative effects here have to be estimated. Another area of study that is technical related to our analysis is the line of work on distributionally robust performative optimization (Jia et al., 2025) and prediction (Xue & Sun, 2024) (see also Section 5). These works use ambiguity sets to robustify optimization and prediction, respectively, against performative distribution shifts. Besides studying generalization instead of prediction or optimization, our work differs in the way ambiguity sets are specified: we estimate them from the sample's performative reactions instead of specifying them *a priori*.

Cutler et al. (2024) analyze stochastic approximation with decision-dependent sampling distributions, but do not study finite-sample generalization from samples to populations under performativity (like the article at hand). Bracale et al.

(2025) develop a framework for micro-foundation inference in strategic prediction, focusing on recovering agents' payoff structures from observed strategic responses rather than on learning-theoretic guarantees. Complementary to both, we derive nonasymptotic generalization and excess-risk bounds for empirical risk minimization when sample and population distributions evolve in response to predictions.

Our work differs from strategic classification (Hardt et al., 2016; Dong et al., 2018; Miller et al., 2020; Sundaram et al., 2023), which assumes agents respond best to predictions with explicit manipulation costs, and from mechanism design (Nisan et al., 2007; Roughgarden, 2016), which assumes rational agents and equilibrium concepts. We instead provide generalization guarantees under minimal structural assumptions—requiring only Lipschitz continuity of the transition map—making our results applicable when the exact nature of performative effects (strategic, behavioral, or intervention-based) is unknown.

The RERM framework has connections to online learning (Cesa-Bianchi & Lugosi, 2006; Hazan, 2016), where models are updated sequentially as new data arrives. However, standard online learning assumes the data distribution is either fixed or evolves independently of the learner's actions, whereas performative prediction creates endogenous nonstationarity. This connects our work to active learning (Settles, 2009; Balcan et al., 2010; Hanneke, 2007; 2014), where the learner adaptively selects which data to observe, and to recent work on learning with feedback loops (Steinke & Zakynthinou, 2020; Chaney et al., 2018). Our bounds reveal that retraining under performativity differs fundamentally from classical nonstationary online learning (Besbes et al., 2015; Russac et al., 2019): the distribution shift is bounded but adversarial (in the sense of creating worst-case generalization gaps), and the learner partially controls this shift through deployment of the model (e.g., through the share $\xi$ of units receiving a treatment, see Section 5). Recent advances in adaptive experimentation (Hadad et al., 2021; Zhan et al., 2021; Zhang et al., 2021) and sequential decision-making under distribution shift (Lu et al., 2021; Chewi et al., 2025) study related phenomena, but typically assume the ability to randomize interventions or maintain a control group—luxuries often unavailable in performative settings where models are deployed system-wide.

Our identification of a fundamental trade-off between changing data and learning from it relates to work on the value of predictions in decision-making (Kleinberg et al., 2015; 2018) and optimal resource allocation under uncertainty (Bastani, 2021). While these works study how prediction quality affects downstream decisions, we show that the act of deploying predictions can degrade future prediction quality through performative effects. This connects to the exploration–exploitation trade-off in bandit problems (Lattimore & Szepesvári, 2020; Slivkins et al., 2019; Rodemann & Augustin, 2021; 2022; 2024; Rodemann et al., 2022; Haltia et al., 2026) and reinforcement learning (Sutton et al., 1998; Kaelbling et al., 1996; Bongratz et al., 2024), where learning requires balancing information gathering with reward maximization. Similarly, recent work on intervention-aware machine learning (Zhang & Bareinboim, 2020) and prediction-informed resource allocation (Kallus, 2018) recognizes that predictive models can serve dual purposes: informing decisions and causing outcomes. Our bounds formalize the cost of this duality, showing that more aggressive interventions (larger $m/n$) improve outcomes but worsen generalization.

# C. Details on Case Study

## C.1. Data Source, Pre-Processing and General Setup

We make use of a real dataset on German job seekers. The data contains information on job-seeker demographics, employment history, and benefits receipt—a rich panel spanning a 2% sample of all administrative labor market records in Germany. Data access was provided via a Scientific Use File from the Research Data Centre (FDZ) of the German Federal Employment Agency (BA) at the Institute for Employment Research (IAB) (Schmucker & vom Berge, 2023b).

**Reproducibility note:** The data are not publicly available due to their sensitive nature. Access can be requested through the Research Data Centre (FDZ) of the Institute for Employment Research (IAB).[9] We provide all preprocessing and analysis code to enable replication for authorized users. For further details on the sampling procedure, feature construction, data sources, and weak anonymization procedures, see the official data report (Schmucker & vom Berge, 2023a).

We construct a prediction task of unemployment duration from this data source, following (Kern et al., 2021). We follow the spell aggregation procedures and feature construction described in Bach et al. (2023); Fischer-Abaigar et al. (2025). We construct covariates capturing demographics (e.g., gender and educational background), information on the last job prior to unemployment, and broader labor market and benefit histories. Some features are frozen at the start of the unemployment spell. Features that evolve mechanically over time (e.g., age and elapsed unemployment duration) are discretized into bins

---

[9]See https://iab.de/en/unit/?id=17.

to obtain a more stable job-seeker cohort and to facilitate the analysis of our bounds. In total, we use 8 ordinal and 20 categorical features. Categorical variables are one-hot encoded, and ordinal variables are mapped to the unit interval $[0, 1]$.

We train a simple logistic regression model in scikit-learn with L2 regularization (regularization strength $C = 1$). Predictive performance is approximately 0.60–0.65 AUC on test sets. Low predictive accuracy is broadly consistent with results in the related literature. Importantly, our objective was not to build the best possible predictor and feature set, but to mimic the modeling choices and data constraints typically faced in employment offices.

### C.1.1. SETUP FOR GENERALIZATION GAP BOUND ON HISTORICAL JOB SEEKER DATA (SEE C.2)

For our analysis of the generalization gap bound computed on historical data (Appendix C.2), we predict over a relatively short time span of 14 days to analyze a fairly stable job-seeker cohort, mimicking an employment office that would continuously retrain and re-predict unemployment outcomes. The interval between model retraining and re-evaluation is also set to 14 days. We further applied a Principal Component Analysis (PCA), reducing the feature set for 28 features to four main features. We focus on a cohort of job seekers who entered unemployment during 2012—we selected this year because it is after major labor market reforms in Germany and before the COVID-19 pandemic. This yields a cohort of roughly 50,000–60,000 individual job seekers, which we further split into training and test sets.

### C.1.2. SETUP FOR BOUND ON SEMI-SIMULATED DATA (SEE C.3)

For the semi-simulation study (Appendix C.3), we focus on individuals 60 days after they entered unemployment to avoid large instabilities at the beginning of spells, as many job seekers find employment very quickly or spend only a short time unemployed. Here, we do not conduct a PCA and work with the full set of 28 features (see below). We set $t = 0$ at the start of the unemployment spell but pre-filter the cohort to individuals who remain unemployed for at least 7 days.

### C.2. Details on the Generalization Gap Bound on Historical Job Seeker Data

As discussed in Section 4, we are using the popular logistic loss $\ell(y, x, \theta) = \log\big(1 + \exp(-y\langle\theta, x\rangle)\big)$, which is $\kappa$-continuously differentiable by Condition 3.3. Its gradient with respect to $\theta$ is given by $\nabla_\theta \ell(y, x, \theta) = -y\,\sigma(-y\langle\theta, x\rangle)x$, where $\sigma(u) = \frac{1}{1+e^{-u}}$. Using $\|\theta\| \le \mathscr{D}_\Theta < \infty$ for all $\theta \in \Theta$ and $\|x\| \le \mathscr{D}_\mathcal{Z} < \infty$ for all $x \in \mathcal{Z}$, so that $|y\langle\theta, x\rangle| \le \mathscr{D}_\mathcal{Z}\mathscr{D}_\Theta$, $\nabla_\theta \ell(y, x, \theta) = -y\,\sigma(-y\langle\theta, x\rangle)x$ is upper bounded by

$$L_\ell = \frac{\mathscr{D}_\mathcal{Z}}{1 + e^{-\mathscr{D}_\mathcal{Z}\mathscr{D}_\Theta}},$$

which proves the loss is $L_\ell$-Lipschitz in $\theta$.

We apply Principal Component Analysis (PCA) to the feature set, resulting in four features, scaled into $[0, 1]$, which (together with $\mathcal{Y} = \{0, 1\}$) gives $\mathcal{D}_\mathcal{Z} = \sqrt{5}$. Moreover, we set $\Theta = [-1000, 1000]^5$, which has a diameter of $\mathcal{D}_\Theta = 2000\sqrt{5}$. Plugging this into $L_\ell = \frac{\mathscr{D}_\mathcal{Z}}{1+e^{-\mathscr{D}_\mathcal{Z}\mathscr{D}_\Theta}}$ gives

$$L_\ell = \frac{\sqrt{5}}{1 + e^{-\sqrt{5}\cdot 2000\sqrt{5}}} = \frac{\sqrt{5}}{1 + e^{-10000}} \approx \sqrt{5} \approx 2.236.$$

We further have (by PCA reduction) $\nu = 4$. We use Wasserstein order $p = 2$ and take $C_a = C_b = 1$ (Fournier & Guillin, 2015). As explained in Section 4 and C.1, we employ L2-regularization with regularization strenght $C = 1$, rendering the loss $\gamma$-strongly convex with $\gamma = \frac{1}{C}$, i.e.,

$$\mathscr{R}(d, \theta) = \mathbb{E}_d[\ell(\theta; z)] + \frac{\lambda}{2}\|\theta\|^2.$$

A standard sensitivity bound for strongly convex objectives gives

$$\|G(d) - G(d')\| \le \frac{1}{\gamma} \sup_\theta \big\|\nabla_\theta\mathscr{R}(d, \theta) - \nabla_\theta\mathscr{R}(d', \theta)\big\|.$$

Moreover, for logistic regression, the per-example gradient is bounded by the feature norm:

$$\|\nabla_\theta\ell(\theta; (x, y))\| \le \|x\| \le D_\mathcal{X} = \sqrt{4} = 2,$$

so a conservative Lipschitz constant of $G$ is

$$L_a \lesssim \frac{D_\mathcal{X}}{\gamma} = \frac{2}{1} = 2.$$

Now recall Corollary 3.8, which states that, in the setting of Theorem 3.7 (Conditions 3.1–3.3), we have

$$\mathscr{R}(d_0, \widehat{\theta}_T) - \mathscr{R}(\widehat{d}_{T-1}, \widehat{\theta}_T) \leq L_\ell \left( \frac{\log(C_a/\delta)}{C_b n} \right)^{\frac{1}{\nu}} + \frac{\varepsilon^T (1 + L_a)^T - 1}{\varepsilon(1 + L_a) - 1} \left( \frac{m}{n} \right)^{\frac{1}{p}} \mathscr{D}_\mathcal{Z},$$

with probability over $d_0$ of at least $1 - 2\delta$.

The first summand is the sampling term. In our setup, using $n = 60147$, $m = 1816$, $L_\ell = 2.236$, $\mathscr{D}_\mathcal{Z} = \sqrt{5}$ with $p = 2$, $\nu = 4$, $C_a = C_b = 1$, $\delta = 0.025$, and $T = 2$, the sampling term becomes

$$L_\ell \left( \frac{\log(1/\delta)}{n} \right)^{1/2} = 2.236 \sqrt{\frac{\log 40}{60147}} \approx 0.017511.$$

Further, using $\varepsilon = m/n$ and $L_\ell^{-1} = 1/2.236$ as well as $\mathscr{D}_\mathcal{Z} = \sqrt{5}$, the second summand simplifies for $T = 2$ to

$$\frac{\varepsilon^T (1 + L_a)^T - 1}{\varepsilon(1 + L_a) - 1} \left( \frac{m}{n} \right)^{\frac{1}{p}} \mathscr{D}_\mathcal{Z} = \frac{5}{3} \sqrt{\frac{m}{n}}.$$

Plugging in $m = 1816$ and $n = 60147$,

$$\frac{5}{3} \sqrt{\frac{1816}{60147}} \approx 0.289601.$$

Therefore, with probability at least $1 - 2\delta = 0.95$,

$$\mathscr{R}(d_0, \widehat{\theta}_2) - \mathscr{R}(\widehat{d}_1, \widehat{\theta}_2) \leq 0.017511 + 0.289601 = 0.307112.$$

### C.3. Details on the Semi-Simulation Study on Job Seeker Data

As detailed in Section 4, we train a binary logistic regression model using, just like detailed in Appendix C.2, the classic logistic loss $\ell(y, x, \theta) = \log(1 + \exp(-y\langle\theta, x\rangle))$, which is $\kappa$-continuously differentiable by Condition 3.3. Its gradient with respect to $\theta$ is given by $\nabla_\theta \ell(y, x, \theta) = -y\,\sigma(-y\langle\theta, x\rangle)x$, where $\sigma(u) = \frac{1}{1+e^{-u}}$. Using $\|\theta\| \leq \mathscr{D}_\Theta < \infty$ for all $\theta \in \Theta$ and $\|x\| \leq \mathscr{D}_\mathcal{Z} < \infty$ for all $x \in \mathcal{Z}$, so that $|y\langle\theta, x\rangle| \leq \mathscr{D}_\mathcal{Z} \mathscr{D}_\Theta$, $\nabla_\theta \ell(y, x, \theta) = -y\,\sigma(-y\langle\theta, x\rangle)x$ is upper bounded by $L_\ell = \frac{\mathscr{D}_\mathcal{Z}}{1+e^{-\mathscr{D}_\mathcal{Z}\mathscr{D}_\Theta}}$, which proves the loss is $L_\ell$-Lipschitz in $\theta$. Again, we use $p$-Wasserstein distance with $p = 2$.

Different to Appendix C.2, we do not use PCA to reduce feature set dimension. Instead, we work with the full set of 28 features as decscribed in Section 4 and Appendix C.1. After normalizing (see Appendix C.1), this yields:

$$\begin{aligned}
\mathcal{D}_\mathcal{Z} &= \sqrt{(1-0)^2 + (1-0)^2 + \cdots + (1-0)^2} \\
&= \sqrt{\underbrace{1 + 1 + \cdots + 1}_{28 \text{ times}}} \\
&= \sqrt{28} \\
&= 2\sqrt{7}.
\end{aligned}$$

So the diameter is $\mathcal{D}_\mathcal{Z} = \sqrt{28} = 2\sqrt{7} \approx 5.29$.

We store predicted probabilities on the train and test sets in CSV files: `train_predictions.csv` and `test_predictions.csv`. Each file contains two columns: the true label $y_i \in \{0, 1\}$ and the predicted probability $\hat{p}_i \in (0, 1)$.

We use confidence level $\delta = 0.05$ and yielding a normal quantile of

$$q(\delta) = \Phi^{-1}(1 - \delta/2) \approx 1.96.$$

Furthermore, we use a normalized, compact parameter set $\Theta = \{\theta : \|\theta\|_2 \leq 1\}$. For logistic regression predictions $f_\theta(x) = \sigma(\theta^\top x)$, since $\sigma'(t) \leq 1/4$,

$$\|\nabla_x f_\theta(x)\|_2 \leq \frac{1}{4}\|\theta\|_2 \leq \frac{1}{4},$$

we get

$$L_f = \frac{1}{4}.$$

As in the case of historical data (see Appendix C.2), the loss Lipschitz constant is approximately equal to $\mathcal{D}_{\mathcal{Z}}$. That is,

$$L_\ell \approx \mathcal{D}_{\mathcal{Z}} = \sqrt{28},$$

and the uniform bound for the prediction range is

$$F = 1,$$

since $f_\theta(x) \in [0, 1]$.

We further use the integral complexity term (as defined in Section 3) denoted by $C_\infty(\mathcal{F})$. For our logistic regression class with compact $\Theta$ and the chosen normalization, we use the numerical value

$$C_\infty(\mathcal{F}) \approx 7.3855,$$

which results from upper bounding $\log N(\mathcal{F}, \|\cdot\|_\infty, \varepsilon)$ using a Lipschitz-in-parameter covering argument and integrate $C_\infty(\mathcal{F}) = \int_0^1 \sqrt{\log N(\mathcal{F}, \|\cdot\|_\infty, \varepsilon)}\, d\varepsilon$.

Theorems 3.13 and 3.15 define the auxiliary radius

$$R(\xi) = \max\left\{\left(q(\delta)\sqrt{\frac{\xi(1-\xi)}{n}}\right)^{1/p} D_Z, \ \xi^{1/p} D_Z\right\}, \qquad p = 2,$$

with $\xi \in \{0.01, \ldots, 0.5\}$ from Section 4.

We compute two bounds, from Theorem 3.13 as well as from Theorem 3.15.

Theorem 3.13 bounds the excess risk under Assumption 3.12 (which is fulfilled by logistic regression, see Section 4) by

$$\frac{48}{\sqrt{n}}\left(C_\infty(\mathcal{F}) + L_\ell L_f\, R(r)^{1-p} D_Z^p\right) + F\sqrt{\frac{2\log(2/\delta)}{n}} + 2L_\ell R(\xi).$$

We decompose the total bound its three summands:

$$\textbf{Complexity term:} \quad \mathsf{Comp}(r) = \frac{48}{\sqrt{n}}\left(C_\infty(\mathcal{F}) + L_\ell L_f/R(\xi)\right), \tag{1}$$

$$\textbf{Sampling term:} \quad \mathsf{Samp}(\xi) = F\sqrt{\frac{2\log(2/\delta)}{n}}, \tag{2}$$

$$\textbf{Performative term:} \quad \mathsf{Perf}(\xi) = 2L_\ell R(r), \tag{3}$$

$$\textbf{Total bound:} \quad \mathsf{Comp}(r) + \mathsf{Samp}(r) + \mathsf{Perf}(r). \tag{4}$$

Code to compute and plot these terms for datasets with varying $\xi$ can be found at https://github.com/rodemann/plt-jobseekers.

Theorem 3.15 is a variant of Theorem 3.13 that replaces the $L_\ell L_f R^{1-p} D_Z^p$ term by a Condition 3.13 constant $B$:

$$\frac{48}{\sqrt{n}}\left(C_\infty(\mathcal{F}) + B\, 2^{p-1}\left(1 + \frac{D_Z}{R(r)}\right)^p\right) + F\sqrt{\frac{2\log(2/\delta)}{n}} + 2L_\ell R(\xi).$$

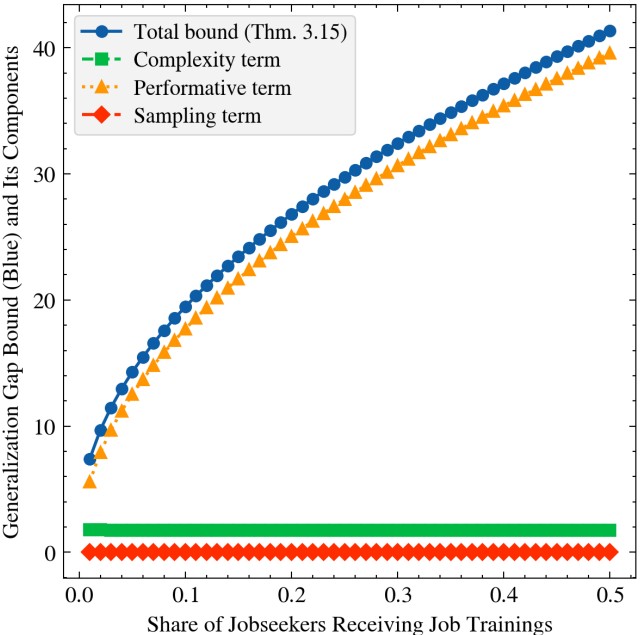

*Figure 2.* Generalization gap bound from Theorem 3.15 and its decomposition as a function of $\xi$ (the fraction of job seekers receiving training). The total bound (blue) is decomposed into complexity term $\mathsf{Comp}_B(\xi)$, sampling term $\mathsf{Samp}(\xi)$, and performative term $\mathsf{Perf}(\xi)$. All values are in logistic loss units. Parameters: $n = 41585$ training samples, $B = 10^{-3}, \delta = 0.05, p = 2$.

We compute this bound exemplarily for varying $\xi$ and $B = 10^{-3}$.

As in our above analysis of Theorem 3.13, we analogously decompose this bound into three terms:

$$\mathsf{Comp}_B(r) + \mathsf{Samp}(r) + \mathsf{Perf}(\xi), \tag{5}$$

changing only

$$\mathsf{Comp}_B(\xi) = \frac{48}{\sqrt{n}}\Big(C_\infty(\mathcal{F}) + 2B(1 + 1/R(\xi))^2\Big). \tag{6}$$

## C.4. Further Results from the Semi-Simulation Study on Job Seeker Data

While Figure 1.3 showing the bound from Theorem 3.13 for different $\xi = m/n$ is in the main paper, we include a visualization of how the generalization gap bound from Theorem 3.15 behaves for varying $\xi$ in what follows.

Namely, Figure 2 visualizes the bound from Theorem 3.15:

$$\frac{48}{\sqrt{n}}\Big(C_\infty(\mathcal{F}) + B\,2^{p-1}\Big(1 + \frac{D_Z}{R(\xi)}\Big)^p\Big) + F\sqrt{\frac{2\log(2/\delta)}{n}} + 2L_\ell R(\xi),$$

which we decompose in the previous section into three components:

$$\mathsf{Comp}_B(r) + \mathsf{Samp}(r) + \mathsf{Perf}(\xi). \tag{7}$$

As an illustration of our results, we compute this bound for varying $\xi$ and $B = 10^{-3}$ as detailed above.

For each $\xi$ (the share of job seekers receiving training), we plot the three components and the total bound. The x-axis is the sweep parameter $\xi = m/n$. The y-axis is the generalization-gap bound (and its components), all in logistic-loss units. We can see that the performative term is dominated by $\frac{48}{\sqrt{n}}C_\infty(\mathcal{F})$ due to our choice of $B$. Code to compute those terms for datasets with varying $\xi$ and plot them can be found in https://github.com/rodemann/plt-jobseekers.

# D. Proofs

### D.1. Proof of Lemma 3.5

We restate the result for ease of exposition.

**Lemma** (In-Sample Performative Shift Bound). *Assume that at most $m$ units (in the sample of size $n$) change in response to predictions at each iteration $t$. If Conditions 3.1–3.3 hold, we have*

$$W_p(\widehat{d}_0, \widehat{d}_T) \leq \frac{[\varepsilon(1 + L_a)]^T - 1}{\varepsilon(1 + L_a) - 1} \left(\frac{m}{n}\right)^{\frac{1}{p}} \mathscr{D}_{\mathcal{Z}},$$

*pointwise in $\widehat{d}_0$ (i.e., for any fixed $\widehat{d}_0$).*

*Proof.* We start by showing that under Conditions 3.1 and 3.3,

$$G : \Delta \to \Theta, \qquad G(d) = \arg\min_{\theta \in \Theta} R(d, \theta)$$

is Lipschitz with respect to $W_p$. This fact is also used in Brown et al. (2022) as Lemma 2 and in Appendix D of Perdomo et al. (2020). For the sake of completeness, we give an independent proof in the following:

Condition 3.3 implies that for every fixed $\theta \in \Theta$, the map

$$z \mapsto \nabla_\theta \ell(z, \theta)$$

is $\kappa$-Lipschitz. Throughout this argument, we use that $\Theta$ is convex. This ensures that the constrained minimizers $G(d)$ and $G(d')$ satisfy the usual first-order variational inequalities. Hence, for all $d, d' \in \Delta$,

$$\|\nabla_\theta \mathscr{R}(d, \theta) - \nabla_\theta \mathscr{R}(d', \theta)\|_2 \leq \kappa W_p(d, d').$$

Combining this with $\gamma$-strong convexity of $\mathscr{R}(d, \cdot)$, which is inherited from $\gamma$-strong convexity of the loss (Condition 3.1), yields

$$\|G(d) - G(d')\|_2 \leq \frac{\kappa}{\gamma} W_p(d, d').$$

Thus $G$ is $L_a$-Lipschitz with $L_a = \kappa/\gamma$.

By the triangle inequality,

$$W_p(\widehat{d}_0, \widehat{d}_T) \leq \sum_{t=0}^{T-1} W_p(\widehat{d}_t, \widehat{d}_{t+1}). \tag{8}$$

for any $\widehat{d}_0$. By the joint $(\varepsilon, p)$-sensitivity of Tr (Condition 3.2), we further have

$$W_p(\widehat{d}_{t+1}, \widehat{d}_t) \leq \varepsilon W_p(\widehat{d}_t, \widehat{d}_{t-1}) + \varepsilon\|\widehat{\theta}_{t+1} - \widehat{\theta}_t\|.$$

Using Condition 3.2 and the $L_a$-Lipschitzness of $G$ (see below), we have

$$W_p(\widehat{d}_{t+1}, \widehat{d}_t) \leq \varepsilon \left( W_p(\widehat{d}_t, \widehat{d}_{t-1}) + \|\widehat{\theta}_{t+1} - \widehat{\theta}_t\| \right)$$

$$\leq \varepsilon \left( W_p(\widehat{d}_t, \widehat{d}_{t-1}) + L_a W_p(\widehat{d}_t, \widehat{d}_{t-1})\| \right)$$

$$= \varepsilon(1 + L_a) W_p(\widehat{d}_t, \widehat{d}_{t-1}) \tag{9}$$

$$\leq \varepsilon^2(1 + L_a)^2 W_p(\widehat{d}_{t-1}, \widehat{d}_{t-2}) \tag{10}$$

$$\leq \ldots \leq$$

$$\leq \varepsilon^t(1 + L_a)^t W_p(\widehat{d}_1, \widehat{d}_0) \tag{11}$$

Summing both sides over $t = 0, \ldots, T - 1$ yields

$$\sum_{t=0}^{T-1} W_p(\widehat{d}_{t+1}, \widehat{d}_t) \leq \sum_{t=0}^{T-1} \varepsilon^t(1 + L_a)^t W_p(\widehat{d}_1, \widehat{d}_0) = \frac{\varepsilon^T(1 + L_a)^T - 1}{\varepsilon(1 + L_a) - 1} \left( W_p(\widehat{d}_1, \widehat{d}_0) \right). \tag{12}$$

Further observe that $W_p(\widehat{d}_1, \widehat{d}_0) = W_p(\mathrm{Tr}(\widehat{d}_0, \widehat{\theta}_1), \widehat{d}_0)$. Since Tr changes $m < n$ units in the sample, we can further bound

$$W_p(\widehat{d}_1, \widehat{d}_0) \leq \left(\frac{m}{n}\right)^{\frac{1}{p}} \mathscr{D}_{\mathcal{Z}}, \tag{13}$$

by the definition of the $p$-Wasserstein distance and by the fact that $\mathscr{D}_{\mathcal{Z}} = \sup_{z,z'} d_{\mathcal{Z}}(z, z') < \infty$. Combining Equations 8, 12 and 13, the result follows immediately. $\qquad\square$

### D.2. Proof of Theorem 3.7

We restate the result for ease of exposition.

**Theorem** (Excess Risk Bound, **RQ1**). *The excess risk $\mathscr{R}(d_0, \widehat{\theta}_T) - \inf_{\theta \in \Theta} \mathscr{R}(d_0, \theta)$ of a model $\widehat{\theta}_T$ (re)trained on performative samples $\widehat{d}_0, \ldots, \widehat{d}_{T-1}$, in which at most $m \leq n$ units change in response to predictions, is upper bounded by*

$$L_\ell \left(\frac{\log(C_a/\delta)}{C_b n}\right)^{\frac{1}{\nu}} + L_\ell L_a \frac{[\varepsilon(1 + L_a)]^{T-1} - 1}{\varepsilon(1 + L_a) - 1} \left(\frac{m}{n}\right)^{\frac{1}{p}} \mathscr{D}_{\mathcal{Z}}$$
$$+ \frac{2 F L_\ell}{\sqrt{n}} \left(12 \mathfrak{C}_{L_2}(\mathcal{F}) + \sqrt{2 \ln(1/\delta)}\right),$$

*for any $T \in \mathbb{N}$, with probability over $d_0$ of at least $1 - 2\delta$, under Conditions 3.1–3.3.*

*Proof.* The idea of the proof is as follows: Bound 1. $\mathscr{R}(d, \widehat{\theta}_T) - \mathscr{R}(\widehat{d}_0, \widehat{\theta}_T)$; 2. $\mathscr{R}(\widehat{d}_0, \widehat{\theta}_T) - \mathscr{R}(\widehat{d}_0, \widehat{\theta}_0)$; as well as 3. $\mathscr{R}(\widehat{d}_0, \widehat{\theta}_0) - \inf_{\theta \in \Theta} \mathscr{R}(d, \theta)$ (all with high probability over $d_0$) and then combine via a union bound argument.

1. We need the Kantorovich-Rubinstein Lemma (Kantorovich & Rubinstein, 1958), which states that, for any $K > 0$ and for any probability measures $\mu$ and $\nu$,

$$W_1(\mu, \nu) = \frac{1}{K} \sup_{\|f\|_L \leq K} \mathbb{E}_{x \sim \mu}[f(x)] - \mathbb{E}_{y \sim \nu}[f(y)],$$

where the supremum is over all Lipschitz continuous functions $f$ with Lipschitz constant at most $K$. (Here, and elsewhere, $\|\cdot\|_L$ denotes the Lipschitz norm.)

By taking $f(\cdot)$ to be the loss function $\ell(\cdot, \theta)$, which is $L_\ell$-Lipschitz by Condition 3.3, the Kantorovich-Rubinstein Lemma implies

$$\left|\mathscr{R}(d, \widehat{\theta}_T) - \mathscr{R}(\widehat{d}_0, \widehat{\theta}_T)\right| \leq L_\ell W_1(d, \widehat{d}_0), \tag{14}$$

where $L_\ell$ is the Lipschitz constant of $\ell$.[10]

We have $W_1(d, \widehat{d}_0) \leq W_p(d, \widehat{d}_0)$ for $1 \leq p \leq 2$. In other words, we can bound the risk difference $\mathscr{R}(d, \widehat{\theta}_T) - \mathscr{R}(\widehat{d}_0, \widehat{\theta}_T)$ via the $p$-Wasserstein distance $W_p(d, \widehat{d}_0)$ between their respective distributions $d$ and $\widehat{d}_0$. We thus have to bound $W_p(d, \widehat{d}_0)$.

Further recall Lemma 3.4:

$$W_p(d_0, \widehat{d}_0) \leq \left(\frac{\log(C_a/\delta)}{C_b n}\right)^{1/\nu}, \tag{15}$$

with probability over $\widehat{d}_0$ of at least $1 - \delta$.

Combining Equations 14 and 15 gives

$$\mathscr{R}(d_0, \widehat{\theta}_T) \leq \mathscr{R}(\widehat{d}_0, \widehat{\theta}_T) + L_\ell \left(\frac{\log(C_a/\delta)}{C_b n}\right)^{1/\nu},$$

with probability over $\widehat{d}_0$ of at least $1 - \delta$.

---

[10]There has been an error in this argument in a previous version of this manuscript, see arxiv note.

2. We will now bound $\|\widehat{\theta}_0 - \widehat{\theta}_T\|$, which will subsequently allow us to bound $\mathscr{R}(\widehat{d}_0, \widehat{\theta}_T) - \mathscr{R}(\widehat{d}_0, \widehat{\theta}_0)$ via Lipschitz continuity of the loss in $\theta$.

By the triangle inequality, we get

$$\|\widehat{\theta}_0 - \widehat{\theta}_T\| \leq \sum_{t=0}^{T-1} \|\widehat{\theta}_t - \widehat{\theta}_{t+1}\|.$$

By the $L_a$-Lipschitz continuity of $G$ and Lemma 3.5,

$$\begin{aligned}
\|\widehat{\theta}_T - \widehat{\theta}_0\|_2 &= \|G(\widehat{d}_T) - G(\widehat{d}_0)\|_2 \\
&\leq L_a W_p(\widehat{d}_T, \widehat{d}_0) \\
&\leq L_a \frac{[\varepsilon(1+L_a)]^T - 1}{\varepsilon(1+L_a) - 1} \left(\frac{m}{n}\right)^{1/p} \mathscr{D}_{\mathcal{Z}}.
\end{aligned}$$

Hence, by Lipschitzness of the loss in $\theta$,

$$\mathscr{R}(\widehat{d}_0, \widehat{\theta}_T) - \mathscr{R}(\widehat{d}_0, \widehat{\theta}_0) \leq L_\ell L_a \frac{[\varepsilon(1+L_a)]^T - 1}{\varepsilon(1+L_a) - 1} \left(\frac{m}{n}\right)^{1/p} \mathscr{D}_{\mathcal{Z}}.$$

3. Define $\mathcal{L} := \ell \circ \mathcal{F}$ as the loss class and recall $\sup_{\theta \in \Theta, x \in \mathcal{X}} \|f_\theta(x)\|_2 \leq F < \infty$ with $f_\theta \in \mathcal{F}$ (see Section 3). As $\ell$ is $L_\ell$-Lipschitz (see 1.), we have that $\mathcal{L}$ is uniformly bounded by $2FL_\ell$. Define

$$\Re_n(\mathscr{F}) := \mathbb{E}\left[\sup_{f \in \mathscr{F}} \frac{1}{n} \sum_{i=1}^{n} \mathcal{E}_i f(X_i)\right],$$

as the expected Rademacher average[11] of any function class $\mathscr{F}$ on $\mathcal{X}$ with Rademacher random variables $\mathcal{E}_1, \ldots, \mathcal{E}_n$ independent of $X_1, \ldots, X_n$. It follows with the standard symmetrization argument (see, e.g., Chapter 26 in Shalev-Shwartz & Ben-David (2014)) for any function class $\mathscr{F}$ uniformly bounded by $B$ that

$$\frac{1}{2}\Re_n(\mathscr{F}) - \frac{B}{\sqrt{n}} \leq \mathbb{E}\left[\sup_{f \in \mathscr{F}} \left[\mathbb{E}_{X \sim d}(f(X)) - \mathbb{E}_{X \sim \widehat{d}_0}(f(X))\right]\right] \leq 2\Re_n(\mathscr{F}).$$

Together with McDiamard's inequality (McDiarmid, 1989), we thus have for the standard empirical risk minimizer $\widehat{\theta}_0$ on $\widehat{d}_0 \overset{\text{i.i.d.}}{\sim} d$ and function the class $\mathscr{F} = \mathcal{L}$ with $B = FL_\ell$

$$\mathscr{R}(\widehat{d}_0, \widehat{\theta}_0) - \inf_{\theta \in \Theta} \mathscr{R}(d, \theta) \leq 2\Re_n(\mathcal{L}) + 2FL_\ell \sqrt{\frac{2\ln(1/\delta)}{n}}, \tag{16}$$

with probability over $\widehat{d}_0$ of at least $1 - \delta$. Talagrand's contraction lemma (Talagrand, 1995) states that for $L_\ell$-Lipschitz loss (given by 1. above), we have $\Re_n(\mathcal{L}) \leq L_\ell \Re_n(\mathcal{F})$. We further have that $\Re_n(\mathcal{F}) = F \Re_n(\mathcal{F}/F)$ by definition of $\Re_n$. Hence, $2\Re_n(\mathcal{L}) \leq 2FL_\ell \Re_n(\mathcal{F}/F)$ in Equation 16. Since all $f_\theta \in \mathcal{F}/F$ are upper-bounded by 1, we can use Dudley's theorem (Dudley, 1987) to obtain

$$\Re_n(\mathcal{F}/F) \leq \frac{12}{\sqrt{n}} \sup_P \int_0^1 \sqrt{\log \mathcal{N}\left(\mathcal{F}, \|\cdot\|_{L_2(P)}^2, \varepsilon\right)} d\varepsilon,$$

with $\mathcal{N}$ the covering number from Definition 3.6, where $\|\cdot\|_{L_2(P)}^2$ was defined as $\|f\|_{L_2(P)}^2 = \mathbb{E}_{X \sim P}[f(X)^2]$ for some measure $P$.

Recall $\mathfrak{C}_{L_2}(\mathcal{F}) := \sup_P \int_0^1 \sqrt{\log \mathcal{N}\left(\mathcal{F}, \|\cdot\|_{L_2(P)}^2, \varepsilon\right)} d\varepsilon$. Thus,

$$\mathscr{R}(\widehat{d}_0, \widehat{\theta}_0) - \inf_{\theta \in \Theta} \mathscr{R}(d, \theta) \leq 2FL_\ell \frac{12}{\sqrt{n}} \mathfrak{C}_{L_2}(\mathcal{F}) + 2FL_\ell \sqrt{\frac{2\ln(1/\delta)}{n}},$$

with probability over $\widehat{d}_0$ of at least $1 - \delta$.

---

[11]See, e.g., Bousquet & Elisseeff (2002) and Von Luxburg & Schölkopf (2011).

The claim then follows by 1., 2., 3., together with a union bound argument. □

## D.3. Proof of Corollary 3.8

Once again, we restate the result for ease of exposition.

**Corollary.** *In the setting of Theorem 3.7 (i.e., under Conditions 3.1–3.3)* $\mathscr{R}(d_0, \widehat{\theta}_T) - \mathscr{R}(\widehat{d}_{T-1}, \widehat{\theta}_T)$ *is upper bounded by*

$$L_\ell \left( \left( \frac{\log\left(C_a/\delta\right)}{C_b n} \right)^{\frac{1}{\nu}} + \frac{\varepsilon^T (1 + L_a)^{T-1} - 1}{\varepsilon (1 + L_a) - 1} \left( \frac{m}{n} \right)^{\frac{1}{p}} \mathscr{D}_{\mathcal{Z}} \right)$$

*with probability over $d_0$ of at least $1 - 2\delta$.*

*Proof.* Recall from Condition 3.3 that the loss $\ell(z, \theta)$ is $L_\ell$-Lipschitz in $z$. By the Kantorovich-Rubinstein Lemma (Kantorovich & Rubinstein, 1958), we have

$$|\mathscr{R}(d_0, \widehat{\theta}_T) - \mathscr{R}(\widehat{d}_0, \widehat{\theta}_T)| \leq L_\ell W_1(d_0, \widehat{d}_0),$$

where $L_\ell$ is the Lipschitz constant of $\ell$.

Since $W_1(d_0, \widehat{d}_0) \leq W_p(d_0, \widehat{d}_0)$ for $1 \leq p \leq 2$, we can bound the risk difference via the $p$-Wasserstein distance between the distributions.

By the triangle inequality, we have

$$W_p(d_0, \widehat{d}_0) \leq W_p(d_0, \widehat{d}_0) + W_p(\widehat{d}_0, \widehat{d}_{T-1}).$$

However, since we want to relate $\mathscr{R}(d_0, \widehat{\theta}_T)$ to $\mathscr{R}(\widehat{d}_{T-1}, \widehat{\theta}_T)$, we observe that by the triangle inequality:

$$W_p(d_0, \widehat{d}_{T-1}) \leq W_p(d_0, \widehat{d}_0) + W_p(\widehat{d}_0, \widehat{d}_{T-1}).$$

From Lemma 3.4, we have

$$W_p(d_0, \widehat{d}_0) \leq \left( \frac{\log(C_a/\delta)}{C_b n} \right)^{1/\nu},$$

with probability at least $1 - \delta$ over $\widehat{d}_0$.

From Lemma 3.5, we have

$$W_p(\widehat{d}_0, \widehat{d}_{T-1}) \leq \frac{\varepsilon^T (1 + L_a)^T - 1}{\varepsilon (1 + L_a) - 1} \left( \frac{m}{n} \right)^{\frac{1}{p}} \mathscr{D}_{\mathcal{Z}}$$

Note that for the bound involving $\widehat{d}_{T-1}$ and $\widehat{\theta}_T$, we use the fact that $T - 1$ iterations have occurred to reach $\widehat{d}_{T-1}$, thus the exponent is $T - 1$ rather than $T$ in the geometric series.

Combining these via the Kantorovich-Rubinstein Lemma yields

$$\mathscr{R}(d_0, \widehat{\theta}_T) - \mathscr{R}(\widehat{d}_{T-1}, \widehat{\theta}_T) \leq L_\ell W_p(d_0, \widehat{d}_{T-1})$$
$$\leq L_\ell \left( \frac{\log(C_a/\delta)}{C_b n} \right)^{1/\nu} + L_\ell \frac{\varepsilon^T (1 + L_a)^T - 1}{\varepsilon (1 + L_a) - 1} \left( \frac{m}{n} \right)^{\frac{1}{p}} \mathscr{D}_{\mathcal{Z}},$$

with probability at least $1 - \delta$, as required. □

## D.4. Proof of Lemma 3.9

We restate the result for ease of readability.

**Lemma** (Performative Population Shift Bound). *Assume $s \in [0, 1]$ is the share of units in $d_0$ reacting to predictions (the "performative response rate"). Then*

$$s < \frac{m}{n} + q(\delta)\sqrt{\frac{m}{n^2}\left(1 - \frac{m}{n}\right)},$$

*with probability $1 - \delta$, where $q(\delta)$ is the $(1 - \delta)$-quantile of the standard normal distribution.*

*Proof.* By treating $m/n$ as an estimator of the parameter $s$ of a Bernoulli distribution (modeling whether statistical units react or not), we have by Wald's method (see e.g., Brown et al., 2001) that

$$s < \frac{m}{n} + q(\delta)\sqrt{\frac{\frac{m}{n}\left(1 - \frac{m}{n}\right)}{n}},$$

with probability $1 - \delta$, as required. □

### D.5. Proof of Theorem 3.10

*Proof.* Fix $d_1 = \mathrm{Tr}(d_0, \widehat{\theta}_T)$. Denote the Bayes-optimal model (from the hypothesis class $\mathcal{F}$ induced by $\Theta$) on $d_0$ by $\theta_{0,B} := \inf_{\theta \in \Theta} \mathcal{R}(d_0, \theta)$ and the one on $d_1$ by $\theta_{1,B} := \inf_{\theta \in \Theta} \mathcal{R}(d_1, \theta)$, respectively. Invoking the Kantorovich-Rubinstein Lemma (see proof of Theorem 3.7) again, we have

$$\mathcal{R}(d_0, \theta_{0,B}) - \mathcal{R}(d_1, \theta_{0,B}) \le L_\ell W_1(d_0, d_1), \tag{17}$$

where $L_\ell$ is the Lipschitz constant of $\ell$.

We have by Lemma 3.5 that

$$W_p(\widehat{d}_0, \widehat{d}_T) \le \frac{\varepsilon^T (1 + L_a)^T - 1}{\varepsilon(1 + L_a) - 1} \left(\frac{m}{n}\right)^{\frac{1}{p}} \mathcal{D}_{\mathcal{Z}}.$$

It is easy to see that the same holds for populations $d_0$ and $d_T$:

$$W_p(d_0, d_T) \le \frac{\varepsilon^T (1 + L_a)^T - 1}{\varepsilon(1 + L_a) - 1} s^{\frac{1}{p}} \mathcal{D}_{\mathcal{Z}},$$

where $s \in [0, 1]$ is the share of units in the population reacting to the predictions (i.e., the population analog of $m/n$) . By treating $m/n$ as an estimator of the parameter $s$ of a Bernoulli distribution (modeling whether statistical units react or not), we have by Wald's method (see e.g., Brown et al., 2001) that

$$s < \frac{m}{n} + q(\delta)\sqrt{\frac{\frac{m}{n}\left(1 - \frac{m}{n}\right)}{n}},$$

with probability $1 - \delta$, where $q(\delta) = q_{1-\delta}$ is the $(1 - \delta)$-quantile of a standard normal distribution (see Lemma 3.9). Hence,

$$W_p(d_0, d_T) \le \frac{\varepsilon^T (1 + L_a)^T - 1}{\varepsilon(1 + L_a) - 1} \left(\frac{m}{n} + \frac{q(\delta)\sqrt{m(n-m)}}{n^{3/2}}\right)^{1/p} \mathcal{D}_{\mathcal{Z}}, \tag{18}$$

with probability $1 - \delta$. Combined with Equation 17, this gives, for $T = 1$,

$$\mathcal{R}(d_0, \theta_{0,B}) - \mathcal{R}(d_1, \theta_{0,B}) \le L_\ell \left(\frac{m}{n} + \frac{q(\delta)\sqrt{m(n-m)}}{n^{3/2}}\right)^{1/p} \mathcal{D}_{\mathcal{Z}}, \tag{19}$$

with probability $1 - \delta$.

We will now bound $\|\theta_{0,B} - \theta_{1,B}\|$, which will allow us to bound $\mathcal{R}(d_1, \theta_{0,B}) - \mathcal{R}(d_1, \theta_{1,B})$ via Lipschitz continuity of the loss in $\theta$. In the next step, this bound on $\mathcal{R}(d_1, \theta_{0,B}) - \mathcal{R}(d_1, \theta_{1,B})$ together with the probabilistic bound on $\mathcal{R}(d_0, \theta_{0,B}) - \mathcal{R}(d_1, \theta_{0,B})$ from Equation 19 will yield a high probability bound on $\mathcal{R}(d_0, \theta_{0,B}) - \mathcal{R}(d_1, \theta_{1,B}) := \inf_{\theta \in \Theta} \mathcal{R}(d_0, \theta) - \inf_{\theta \in \Theta} \mathcal{R}(\mathrm{Tr}(d_0, \widehat{\theta}_T), \theta)$. (Recall we fixed $d_1 = \mathrm{Tr}(d_0, \widehat{\theta}_T)$.)

We have $\|\theta_{0,B} - \theta_{1,B}\| \leq L_a W_p(d_1, d_0)$ by Lipschitz continuity of $G$ (see proof of Lemma 3.5). Further applying Lemma 3.9 with $T = 1$ gives

$$\|\theta_{0,B} - \theta_{1,B}\| \leq L_a s^{\frac{1}{p}} \mathscr{D}_{\mathcal{Z}},$$

where $L_a$ is the Lipschitz constant of $\Delta \to \Theta : d \mapsto \arg\min_{\theta \in \Theta} \mathscr{R}(d, \theta)$ as above.

With reasoning analogous to Equation 18, we can upper bound $s$ by our tail distribution of our estimator $m/n$ with probability $1 - \delta$, yielding

$$s < \frac{m}{n} + q(\delta)\sqrt{\frac{\frac{m}{n}(1 - \frac{m}{n})}{n}},$$

with probability $1 - \delta$, where again $q(\delta) = q_{1-\delta}$ is the $(1 - \delta)$-quantile of a standard normal distribution. Hence,

$$\|\theta_{0,B} - \theta_{1,B}\| \leq L_a s^{\frac{1}{p}} \mathscr{D}_{\mathcal{Z}} \leq L_a \left( \frac{m}{n} + \frac{q(\delta)\sqrt{m(n-m)}}{n^{3/2}} \right)^{1/p} \mathscr{D}_{\mathcal{Z}},$$

with probability $1 - \delta$. It follows from the linearity of expectation and the loss being Lipschitz in $\theta$ that

$$\mathscr{R}(d_1, \theta_{0,B}) - \mathscr{R}(d_1, \theta_{1,B}) \leq L_\ell \|\theta_{0,B} - \theta_{1,B}\|.$$

Thus,

$$\mathscr{R}(d_1, \theta_{0,B}) - \mathscr{R}(d_1, \theta_{1,B}) \leq L_\ell L_a \cdot \left( \frac{m}{n} + \frac{q(\delta)\sqrt{m(n-m)}}{n^{3/2}} \right)^{1/p} \mathscr{D}_{\mathcal{Z}},$$

with probability $1 - \delta$.

Together with Equation 19, this gives an upper bound with high probability on $\mathscr{R}(d_0, \theta_{0,B}) - \mathscr{R}(d_1, \theta_{1,B}) := \inf_{\theta \in \Theta} \mathscr{R}(d_0, \theta) - \inf_{\theta \in \Theta} \mathscr{R}(\text{Tr}(d_0, \widehat{\theta}_T), \theta)$. Namely,

$$\inf_{\theta \in \Theta} \mathscr{R}(d_0, \theta) - \inf_{\theta \in \Theta} \mathscr{R}(d_1, \theta) \leq L_\ell L_a \left( \frac{m}{n} + \frac{q(\delta)\sqrt{m(n-m)}}{n^{3/2}} \right)^{1/p} \mathscr{D}_{\mathcal{Z}} + L_\ell \left( \frac{m}{n} + \frac{q(\delta)\sqrt{m(n-m)}}{n^{3/2}} \right)^{1/p} \mathscr{D}_{\mathcal{Z}}, \quad (20)$$

with probability $1 - \delta$. Or equivalently,

$$\inf_{\theta \in \Theta} \mathscr{R}(d_0, \theta) - \inf_{\theta \in \Theta} \mathscr{R}(d_1, \theta) \leq L_\ell \mathscr{D}_{\mathcal{Z}} \left( \frac{m}{n} + \frac{q(\delta)\sqrt{m(n-m)}}{n^{3/2}} \right)^{1/p} (1 + L_a), \quad (21)$$

with probability $1 - \delta$. Analogously to the proof of Theorem 3.7, we can now bound 1. $\mathscr{R}(d_0, \widehat{\theta}_T) - \mathscr{R}(\widehat{d}_0, \widehat{\theta}_T)$, 2. $\mathscr{R}(\widehat{d}_0, \widehat{\theta}_T) - \mathscr{R}(\widehat{d}_0, \widehat{\theta}_0)$, as well as 3. $\mathscr{R}(\widehat{d}_0, \widehat{\theta}_0) - \inf_{\theta \in \Theta} \mathscr{R}(d_0, \theta)$ (all with high probability over $\widehat{d}_0$) and then combine via the union bound to get

$$L_\ell \left( \frac{\log(C_a/\delta)}{C_b n} \right)^{1/\nu} + L_\ell L_a \frac{[\varepsilon(1 + L_a)]^T - 1}{\varepsilon(1 + L_a) - 1} \left( \frac{m}{n} \right)^{1/p} \mathscr{D}_{\mathcal{Z}} + \frac{F L_\ell}{\sqrt{n}} \left( 24 \mathfrak{C}_{L_2}(\mathcal{F}) + 2\sqrt{2\ln(1/\delta)} \right), \quad (22)$$

as an upper bound on $\mathscr{R}(d_0, \widehat{\theta}_T) - \inf_{\theta \in \Theta} \mathscr{R}(d_0, \theta)$ for any $T \in \mathbb{N}$ with probability of at least $1 - 2\delta$ under Conditions 3.1 and 3.2.

Recall we want to bound the *performative excess risk*

$$\mathscr{R}(d_1, \widehat{\theta}_T) - \inf_{\theta \in \Theta} \mathscr{R}(d_1, \theta) = \mathscr{R}(\text{Tr}(d_0, \widehat{\theta}_T), \widehat{\theta}_T) - \inf_{\theta \in \Theta} \mathscr{R}(\text{Tr}(d_0, \widehat{\theta}_T), \theta)$$

of a model $\widehat{\theta}_T$ trained on performative samples $\widehat{d}_0, \ldots, \widehat{d}_T$, in which $m \leq n$ units react to the predictions. Obviously,

$$\mathscr{R}(d_1, \widehat{\theta}_T) - \inf_{\theta \in \Theta} \mathscr{R}(d_1, \theta) = \mathscr{R}(d_1, \widehat{\theta}_T) - \mathscr{R}(d_0, \widehat{\theta}_T) + \mathscr{R}(d_0, \widehat{\theta}_T) - \inf_{\theta \in \Theta} \mathscr{R}(d_0, \theta) + \inf_{\theta \in \Theta} \mathscr{R}(d_0, \theta) - \inf_{\theta \in \Theta} \mathscr{R}(d_1, \theta). \quad (23)$$

We have, by reasoning from Equations 19 and 17 as well as the symmetry of the Wasserstein distance, that

$$\mathscr{R}(d_1, \theta) - \mathscr{R}(d_0, \theta) \leq L_\ell \left( \frac{m}{n} + \frac{q(\delta)\sqrt{m(n-m)}}{n^{3/2}} \right)^{1/p} \mathscr{D}_{\mathcal{Z}},$$

for any $\theta \in \Theta$ with probability $1 - \delta$. Applying this to $\theta_T$ and using Equation 21 as well as Equation 22, we can upper bound our target $\mathscr{R}(d_1, \widehat{\theta}_T) - \inf_{\theta \in \Theta} \mathscr{R}(d_1, \theta)$ (Equation 23) by

$$\underbrace{\left( \frac{m}{n} + \frac{q(\delta)\sqrt{m(n-m)}}{n^{3/2}} \right)^{1/p}}_{\geq \mathscr{R}(d_1,\theta) - \mathscr{R}(d_0,\theta)} + \underbrace{L_\ell \left( \frac{\log(C_a/\delta)}{C_b n} \right)^{\frac{1}{\nu}} + L_\ell L_a \frac{[\varepsilon(1+L_a)]^T - 1}{\varepsilon(1+L_a) - 1} \left( \frac{m}{n} \right)^{1/p} \mathscr{D}_{\mathcal{Z}} + \frac{FL_\ell}{\sqrt{n}} \left( 24\, \mathfrak{C}_{L_2}(\mathcal{F}) + 2\sqrt{2\ln(1/\delta)} \right)}_{\geq \mathscr{R}(d_0,\widehat{\theta}_T) - \inf_{\theta \in \Theta} \mathscr{R}(d_0,\theta)}$$

$$+ \underbrace{(1 + L_a)L_\ell \, \mathscr{D}_{\mathcal{Z}} \left( \frac{m}{n} + \frac{q(\delta)\sqrt{m(n-m)}}{n^{3/2}} \right)^{1/p}}_{\geq \inf_{\theta \in \Theta} \mathscr{R}(d_0,\theta) - \inf_{\theta \in \Theta} \mathscr{R}(d_1,\theta)},$$

for any $T \in \mathbb{N}$ with probability $1 - 4\delta$.

Equivalently, $\mathscr{R}(d_1, \widehat{\theta}_T) - \inf_{\theta \in \Theta} \mathscr{R}(d_1, \theta)$ is upper bounded by

$$L_\ell \left( \frac{\log(C_a/\delta)}{C_b n} \right)^{\frac{1}{\nu}} + L_\ell L_a \frac{[\varepsilon(1+L_a)]^T - 1}{\varepsilon(1+L_a) - 1} \left( \frac{m}{n} \right)^{\frac{1}{p}} \mathscr{D}_{\mathcal{Z}}$$

$$+ \frac{FL_\ell}{\sqrt{n}} \left( 24\, \mathfrak{C}_{L_2}(\mathcal{F}) + 2\sqrt{2\ln(1/\delta)} \right) + (1 + L_\ell + L_a L_\ell) \, \mathscr{D}_{\mathcal{Z}} \left( \frac{m}{n} + \frac{q(\delta)\sqrt{m(n-m)}}{n^{3/2}} \right)^{1/p}.$$

Simplifying further, we obtain

$$\mathscr{R}(d_1, \widehat{\theta}_T) - \inf_{\theta \in \Theta} \mathscr{R}(d_1, \theta) \leq A(m,n) + L_\ell \left[ C(n) + \frac{2F}{\sqrt{n}} \left( 12\mathfrak{C}_{L_2}(\mathcal{F}) + \sqrt{2\ln(1/\delta)} \right) + L_a K(T,m,n) \right],$$

for any $T \in \mathbb{N}$ with probability $1 - 4\delta$, where

$$A(m, n) := (1 + L_\ell + L_a L_\ell) \mathscr{D}_{\mathcal{Z}} \left( \frac{m}{n} + \frac{q(\delta)\sqrt{m(n-m)}}{n^{3/2}} \right)^{1/p},$$

$$C(n) := \left( \frac{\log(C_a/\delta)}{C_b n} \right)^{\frac{1}{\nu}},$$

$$K(T, m, n) := \frac{[\varepsilon(1+L_a)]^T - 1}{\varepsilon(1+L_a) - 1} \left( \frac{m}{n} \right)^{1/p} \mathscr{D}_{\mathcal{Z}},$$

only depend on constants as well as on $m$, $n$ and $T$, as required. $\qquad\square$

### D.6. Proof of Corollary 3.11

We restate the corollary first.

**Corollary** (Improving Bounds Under Performativity).

*I)* $\widehat{\theta}_0$ *yields the tightest performative excess risk bound among* $\{\widehat{\theta}_t\}_{t=0}^T$.
*II)* *It holds with probability* $1 - \delta$ *that*

$$s < \frac{M_t}{nT} + q(\delta)\sqrt{\frac{M_T}{n^2 T^2} \left( 1 - \frac{M_T}{nT} \right)},$$

*where* $M_T = \sum_{t=1}^T m_t$, *leading to an as-tight or tighter performative excess risk bound as the one in Theorem 3.10.*

*Proof.* We prove both parts separately.

**I)** From Theorem 3.10, the $T$-dependent part of the performative excess risk bound is contained in

$$K(T, m, n) = \frac{[\varepsilon(1 + L_a)]^T - 1}{\varepsilon(1 + L_a) - 1} \left(\frac{m}{n}\right)^{1/p} \mathscr{D}_{\mathcal{Z}},$$

with the usual convention that the fraction is $T$ when $\varepsilon(1 + L_a) = 1$. Equivalently,

$$K(T, m, n) = \left(\frac{m}{n}\right)^{1/p} \mathscr{D}_{\mathcal{Z}} \sum_{j=0}^{T-1} [\varepsilon(1 + L_a)]^j.$$

Since $\varepsilon > 0$ and $L_a > 0$, this geometric sum is nondecreasing in $T$. Hence, the bound is minimized by using the initial fit $\widehat{\theta}_0$, i.e. before additional sample-performative retraining rounds.

**II)** In Lemma 3.9, we bounded $s$ (the population performative response rate) using a single observation $m/n$:

$$s < \frac{m}{n} + q(\delta)\sqrt{\frac{m/n(1 - m/n)}{n}}.$$

However, when retraining $T$ times, we observe $m_1, \ldots, m_T$ at each iteration. Define $M_T = \sum_{t=1}^T m_t$. By treating each $m_t/n$ as an independent Bernoulli estimate of $s$ and pooling across iterations, we obtain $M_T/(Tn)$ as an improved estimator of $s$ based on effective sample size $Tn$.

Applying Wald's method to this estimator:

$$s < \frac{M_t}{Tn} + q(\delta)\sqrt{\frac{M_T/(Tn)(1 - M_T/(Tn))}{Tn}} = \frac{M_t}{Tn} + q(\delta)\sqrt{\frac{M_T(Tn - M_T)}{T^2 n^2}}.$$

Since $M_T \leq \max\{m_1, \ldots, m_T\}T = mT$ and the variance term decreases with larger sample size, this bound is at least as tight as (and typically tighter than) the single-iteration bound from Lemma 3.9.

Substituting this improved bound into Theorem 3.10 yields a tighter performative excess risk bound. $\qquad\square$

## D.7. Proof of Theorem 3.13

We restate the theorem for ease of exposition.

**Theorem** (Generalization Gap I, **RQ2**). *Under Conditions 3.1–3.3 and 3.12, the performative generalization gap* $\mathscr{R}(\mathrm{Tr}(d_0, \widehat{\theta}_T), \widehat{\theta}_T) - \mathscr{R}(\mathrm{Tr}(\widehat{d}_0, \widehat{\theta}_T), \widehat{\theta}_T)$ *is upper bounded with probability* $1 - 3\delta$ *over* $d_0$ *by*

$$\frac{48}{\sqrt{n}} \left(\mathfrak{C}_{\infty(d_0)}(\mathcal{F}) + \frac{L_\ell L_f \mathscr{D}_{\mathcal{Z}}{}^p}{R^{p-1}}\right) + F\sqrt{\frac{2\log(2/\delta)}{n}} + 2L_\ell R,$$

*where* $R := \max\left\{ \left(q(\delta)\sqrt{\frac{\frac{m}{n}(1 - \frac{m}{n})}{n}}\right)^{\frac{1}{p}} \mathscr{D}_{\mathcal{Z}}, \; \frac{\varepsilon^T(1 + L_a)^T - 1}{\varepsilon(1 + L_a) - 1} \left(\frac{m}{n}\right)^{\frac{1}{p}} \mathscr{D}_{\mathcal{Z}} \right\}.$

*Proof.* We want to bound the generalization gap under performativity $\mathscr{R}(\mathrm{Tr}(d_0, \widehat{\theta}_T), \widehat{\theta}_T) - \mathscr{R}(\mathrm{Tr}(\widehat{d}_0, \widehat{\theta}_T), \widehat{\theta}_T)$. Recall from Lemma 3.5,

$$W_p(\widehat{d}_0, \widehat{d}_T) \leq L_\ell \frac{\varepsilon^T(1 + L_a)^T - 1}{\varepsilon(1 + L_a) - 1} \left(\frac{m}{n}\right)^{\frac{1}{p}} \mathscr{D}_{\mathcal{Z}},$$

and analogously for the population,

$$W_p(d_0, d_T) \leq \frac{\varepsilon^T(1 + L_a)^T - 1}{\varepsilon(1 + L_a) - 1} (s)^{\frac{1}{p}} \mathscr{D}_{\mathcal{Z}}.$$

Slightly overloading notation, denote $d_1 = \text{Tr}(d_0, \widehat{\theta}_T)$ and $\widehat{d}_1 = \text{Tr}(\widehat{d}_0, \widehat{\theta}_T)$, respectively. We have $W_p(d_0, d_1) \leq s^{1/p} \mathscr{D}_{\mathcal{Z}}$ and $W_p(\widehat{d}_0, \widehat{d}_1) \leq \left(\frac{m}{n}\right)^{1/p} \mathscr{D}_{\mathcal{Z}}$. This implies

$$\mathscr{R}(\text{Tr}(d_0, \widehat{\theta}_T), \widehat{\theta}_T) - \mathscr{R}(\text{Tr}(\widehat{d}_0, \widehat{\theta}_T), \widehat{\theta}_T) \leq \sup_{d: W_p(d_0, d) \leq s^{\frac{1}{p}} \mathscr{D}_{\mathcal{Z}}} \mathscr{R}(d, \widehat{\theta}_T) - \inf_{d: W_p(\widehat{d}_0, d) \leq \left(\frac{m}{n}\right)^{\frac{1}{p}} \mathscr{D}_{\mathcal{Z}}} \mathscr{R}(d, \widehat{\theta}_T).$$

Note that by reasoning from the proof of Theorem 3.10 we have that

$$s < \frac{m}{n} + q(\delta)\sqrt{\frac{\frac{m}{n}\left(1 - \frac{m}{n}\right)}{n}},$$

with probability $1 - \delta$, where $q(\delta) = q_{1-\delta}$ is the $(1 - \delta)$-quantile of a standard normal distribution.

Informed by this result, choose a common radius $R := \max\left\{ \left(\frac{m}{n} + q(\delta)\sqrt{\frac{\frac{m}{n}\left(1 - \frac{m}{n}\right)}{n}}\right)^{\frac{1}{p}} \mathscr{D}_{\mathcal{Z}}, \; \frac{\varepsilon^T(1+L_a)^T - 1}{\varepsilon(1+L_a) - 1} \left(\frac{m}{n}\right)^{\frac{1}{p}} \mathscr{D}_{\mathcal{Z}} \right\}.$

Then trivially

$$\{d : W_p(\widehat{d}_0, d) \leq r_2\} \subseteq \{d : W_p(\widehat{d}_0, d) \leq R\}.$$

It also holds with probability $1 - \delta$ that

$$\{d : W_p(d_0, d) \leq r_1\} \subseteq \{d : W_p(d_0, d) \leq R\}.$$

Therefore,

$$\begin{aligned}
\mathscr{R}(\text{Tr}(d_0, &\widehat{\theta}_T), \widehat{\theta}_T) - \mathscr{R}(\text{Tr}(\widehat{d}_0, \widehat{\theta}_T), \widehat{\theta}_T) \\
&\leq \sup_{W_p(d_0, d) \leq R} \mathscr{R}(d, \widehat{\theta}_T) - \inf_{W_p(\widehat{d}_0, d) \leq R} \mathscr{R}(d, \widehat{\theta}_T),
\end{aligned} \tag{24}$$

with probability $1 - \delta$. The right-hand side of the previous equation is equal to

$$\underbrace{\sup_{W_p(d_0, d) \leq R} \mathscr{R}(d, \widehat{\theta}_T) - \sup_{W_p(\widehat{d}_0, d) \leq R} \mathscr{R}(d, \widehat{\theta}_T)}_{(a)} + \underbrace{\sup_{W_p(\widehat{d}_0, d) \leq R} \mathscr{R}(d, \widehat{\theta}_T) - \inf_{W_p(\widehat{d}_0, d) \leq R} \mathscr{R}(d, \widehat{\theta}_T)}_{(b)}$$

We will first bound (a), then (b). In (a), we are looking at two supremum risk functionals with respect to two Wasserstein balls with same radius $R$, but with different centers $\widehat{d}_0$ and $d_0$. In (b), we are looking at two infimum/supremum risk functionals with respect to the same Wasserstein ball (same radius $R$ and same center $\widehat{d}_0$).

(a) Observe that $\sup_{W_p(d_0, d) \leq R} \mathscr{R}(d, \widehat{\theta}_T)$ and $\sup_{W_p(\widehat{d}_0, d) \leq R} \mathscr{R}(d, \widehat{\theta}_T)$ are distributionally robust risk functionals. We can thus use the following dual characterization of a distributionally robust risk from Gao & Kleywegt (2023) for supremum risks centered around some $d_c$:

$$\sup_{W_p(d, d_c) \leq R} \mathscr{R}(d, \theta) = \min_{\lambda \geq 0} \left\{ \lambda R^p + \mathbb{E}_{d_c}\left[\varphi_{\lambda, \theta}(Z)\right] \right\}, \tag{25}$$

where

$$\varphi_{\lambda, \theta}(z) := \sup_{z' \in \mathcal{Z}} \left\{ \ell(f_\theta(x'), y') - \lambda d_{\mathcal{Z}}^p(z, z') \right\},$$

for any $\theta \in \Theta$ and any $\lambda \geq 0$ and $d_{\mathcal{Z}}$ a metric on $\mathcal{Z}$. Gao & Kleywegt (2023) prove this characterization holds for any upper semi-continuous function (including, importantly for our case, $\ell \circ f_\theta : \mathcal{Z} \to \mathbb{R}$) and for any $d$ with $p$-finite moments, which holds in our setup with $1 \leq p \leq 2$. Since $f_\theta$ is continuous by Condition 3.12 and $\ell$ is continuous in the data by Condition 3.3, we can apply this result to (a) with respect to both true law $d_0$ and sample $\widehat{d}_0$ and obtain

$$\sup_{W_p(d_0, d) \leq R} \mathscr{R}(d, \widehat{\theta}_T) - \sup_{W_p(\widehat{d}_0, d) \leq R} \mathscr{R}(d, \widehat{\theta}_T) = \min_{\lambda \geq 0}\left\{ \lambda R^p + \mathbb{E}_{d_0}\left[\varphi_{\lambda, \widehat{\theta}_T}(Z)\right] \right\} - \min_{\lambda \geq 0}\left\{ \lambda R^p + \mathbb{E}_{\widehat{d}_0}\left[\varphi_{\lambda, \widehat{\theta}_T}(Z)\right] \right\}. \tag{26}$$

Defining $\widehat{\lambda} := \arg\min_{\lambda} \left\{ \lambda R^p + \mathbb{E}_{\widehat{d}_0}\left[\varphi_{\lambda,\widehat{\theta}_T}(Z)\right] \right\}$, we get

$$
\begin{aligned}
\min_{\lambda \geq 0} & \left\{ \lambda R^p + \int_z \varphi_{\lambda,\widehat{\theta}_T}(z)\, d_0(\mathrm{d}z) \right\} - \min_{\lambda \geq 0} \left\{ \lambda R^p + \int_z \varphi_{\lambda,\widehat{\theta}_T}(z)\, \widehat{d}_0(\mathrm{d}z) \right\} \\
&= \min_{\lambda \geq 0} \left\{ \lambda R^p + \int_z \varphi_{\lambda,\widehat{\theta}_T}(z)\, d_0(\mathrm{d}z) \right\} - \left( \widehat{\lambda} R^p + \int_z \varphi_{\widehat{\lambda},\widehat{\theta}_T}(z)\, \widehat{d}_0(\mathrm{d}z) \right) \\
&\leq \int_z \varphi_{\widehat{\lambda},\widehat{\theta}_T}(z)\,(d_0 - \widehat{d}_0)(\mathrm{d}z).
\end{aligned}
\tag{27}
$$

We will now show—closely following Lemma 1 of Lee & Raginsky (2018)—that $\widehat{\lambda} \leq L_\ell L_f R^{1-p}$, where $L_\ell L_f$ is the Lipschitz constant of $\ell \circ f_\theta$ as above.

Clearly,

$$
\widehat{\lambda} R^p \leq \widehat{\lambda} R^p + \mathbb{E}_{\widehat{d}_0}\left[ \sup_{z' \in \mathcal{Z}} \left\{ f_{\widehat{\theta}_T} \circ \ell(z') - f_{\widehat{\theta}_T} \circ \ell(Z) - \widehat{\lambda}\, d_z^p(z, z') \right\} \right].
$$

Since $\widehat{\lambda}$ is optimal with respect to $\widehat{\theta}_T$ (see definition above) and due to $\ell \circ f_\theta$ being $L_\ell L_f$-Lipschitz we have that, for all $\lambda \geq 0$,

$$
\begin{aligned}
\widehat{\lambda} R^p &+ \mathbb{E}_{\widehat{d}_0}\left[ \sup_{z' \in \mathcal{Z}} \left\{ f_{\widehat{\theta}_T} \circ \ell(z') - f_{\widehat{\theta}_T} \circ \ell(Z) - \widehat{\lambda} d_{\mathcal{Z}}^p(z, z') \right\} \right] \\
&\leq \lambda R^p + \mathbb{E}_{\widehat{d}_0}\left[ \sup_{z' \in \mathcal{Z}} \left\{ f_{\widehat{\theta}_T} \circ \ell(z') - f_{\widehat{\theta}_T} \circ \ell(Z) - \lambda d_{\mathcal{Z}}^p(z, z') \right\} \right] \\
&\leq \lambda R^p + \mathbb{E}_{\widehat{d}_0}\left[ \sup_{z' \in \mathcal{Z}} \left\{ L_\ell L_f d_{\mathcal{Z}}(z, z') - \lambda d_{\mathcal{Z}}^p(z, z') \right\} \right] \\
&\leq \lambda R^p + \sup_{t \geq 0} \left\{ L_\ell L_f t - \lambda t^p \right\},
\end{aligned}
$$

where we parametrize $t = d_{\mathcal{Z}}(z, z')$ as in Lee & Raginsky (2018). If $p = 1$, we have

$$
\widehat{\lambda} R \leq L_\ell L_f R + \sup_{t \geq 0}\{ L_\ell L_f t - L_\ell L_f t \} = L_\ell L_f R,
$$

by setting $L_\ell L_f = \lambda$, which gives $\widehat{\lambda} \leq L_\ell L_f$. In the case of $p > 1$, we have

$$
\widehat{\lambda} R \leq \lambda R^p + L_\ell L_f^{\frac{p}{p-1}} p^{-\frac{p}{p-1}} (p-1) \lambda^{-\frac{1}{p-1}},
$$

via the fact that $\arg\sup_{t \geq 0}\{ L_\ell L_f t - L_\ell L_f t \} = (L_\ell L_f / p\lambda)^{1/(p-1)}$. Minimizing the right-hand side over $\lambda \geq 0$ with the choice of $\lambda = L_\ell L_f / p R^{p-1}$, we get the above stated bound:

$$
\widehat{\lambda} \leq L_\ell L_f R R^{-p}.
$$

In analogy to Theorem 2 of Lee & Raginsky (2018), we can now define the function class

$$
\Phi := \{ \varphi_{\lambda,\theta} : \lambda \leq L_\ell L_f R^{1-p}, f_\theta \in \mathcal{F} \},
$$

such that

$$
\sup_{W_p(d_0, d) \leq R} \mathscr{R}(d, \widehat{\theta}_T) - \sup_{W_p(\widehat{d}_0, d) \leq R} \mathscr{R}(d, \widehat{\theta}_T) \leq \int_z \varphi_{\widehat{\lambda},\widehat{\theta}_T}(z)\,(d_0 - \widehat{d}_0)(\mathrm{d}z) \leq \sup_{\varphi \in \Phi} \int_{\mathcal{Z}} \varphi(Z)\,(d_0 - \widehat{d}_0),
$$

where we used Equations 26 and 27.

We can eventually obtain the following Rademacher bound by standard symmetrization, following the proof of Theorem 3 in Lee & Raginsky (2018):

$$
\sup_{\varphi \in \Phi} \int_{\mathcal{Z}} \varphi(Z)\,(d_0 - \widehat{d}_0) \leq 2\Re_n(\Phi) + F\sqrt{\frac{2\log(2/\delta)}{n}},
$$

with probability of at least $1 - 2\delta$. Here $F$ is the upper bound of all $f_\theta \in \mathcal{F}$ (see Section 3) and

$$\Re_n(\Phi) := \mathbb{E}\left[\sup_{\varphi \in \Phi} \frac{1}{n} \sum_{i=1}^{n} \varepsilon_i \varphi(Z_i)\right]$$

is the standard empirical Rademacher average (see, e.g., Chapter 26 in Shalev-Shwartz & Ben-David 2014, Bousquet & Elisseeff 2002 or Von Luxburg & Schölkopf 2011) of the above defined function class $\Phi$ with Rademacher random variables $\varepsilon_1, \ldots, \varepsilon_n$, independent of the random variables describing the data.

All in all, we obtain for Part (a):

$$\sup_{W_p(d_0,d) \leq R} \mathscr{R}(d, \widehat{\theta}_T) - \sup_{W_p(\widehat{d}_0, d) \leq R} \mathscr{R}(d, \widehat{\theta}_T) \leq 2\Re_n(\Phi) + F\sqrt{\frac{2\log(2/\delta)}{n}}.$$

(b) Recall the classic Kantorovich-Rubinstein Lemma (see the proof of Theorem 3.7):

$$\mathscr{R}(d', \theta) - \mathscr{R}(d'', \theta) \leq L_\ell W_1(d', d''),$$

for any $\theta$ and any $d', d''$, where $L_\ell$ is the Lipschitz constant of $\ell$. Also, it trivially holds that for any $d', d''$ such that $W_p(\widehat{d}_0, d') \leq R$ and $W_p(\widehat{d}_0, d'') \leq R$ (i.e., for any $d', d''$ in the Wasserstein ball around $\widehat{d}_0$ of radius $R$) that $W_p(d', d'') \leq 2R$. This directly implies that

$$\sup_{W_p(\widehat{d}_0, d) \leq R} \mathscr{R}(d, \widehat{\theta}_T) - \inf_{W_p(\widehat{d}_0, d) \leq R} \mathscr{R}(d, \widehat{\theta}_T) \leq 2L_\ell R.$$

Combining Parts (a) and (b), we conclude that $\mathscr{R}(\mathrm{Tr}(d_0, \widehat{\theta}_T), \widehat{\theta}_T) - \mathscr{R}(\mathrm{Tr}(\widehat{d}_0, \widehat{\theta}_T), \widehat{\theta}_T)$ is at most equal to

$$\underbrace{\sup_{W_p(d_0,d) \leq R} \mathscr{R}(d, \widehat{\theta}_T) - \sup_{W_p(\widehat{d}_0, d) \leq R} \mathscr{R}(d, \widehat{\theta}_T)}_{\leq 2\Re_n(\Phi) + F\sqrt{\frac{2\log(2/\delta)}{n}}} + \underbrace{\sup_{W_p(\widehat{d}_0, d) \leq R} \mathscr{R}(d, \widehat{\theta}_T) - \inf_{W_p(\widehat{d}_0, d) \leq R} \mathscr{R}(d, \widehat{\theta}_T)}_{\leq 2L_\ell R}, \tag{28}$$

with probability $1 - 2\delta$. All that is left to do now is to bound the Rademacher average $\Re_n(\Phi)$ of the above defined function class $\Phi := \{\varphi_{\lambda,\theta} : \lambda \leq L_\ell L_f R^{1-p}, f_\theta \in \mathcal{F}\}$. To this end, define the $\Phi$-indexed process $X = (X_\varphi)_{\varphi \in \Phi}$ via $X_\varphi := \frac{1}{\sqrt{n}} \sum_{i=1}^{n} \varepsilon_i \varphi(Z_i)$, with $\mathbb{E}[X_\varphi] = 0$ for all $\varphi \in \Phi$. Following Lemma 5 of Lee & Raginsky (2018), we first show that $X$ is a subgaussian process with respect to a suitable pseudometric.

For $\varphi = \varphi_{\lambda,f}$ and $\varphi' = \varphi_{\lambda',f'}$, define $d_\Phi(\varphi, \varphi') := \|f - f'\|_{L_\infty(d_0)} + \mathscr{D}_{\mathcal{Z}}^p |\lambda - \lambda'|$, for which it is easy to see that $\|\varphi - \varphi'\|_{L_\infty(d_0)} \leq d_\Phi(\varphi, \varphi')$. Using Hoeffding's lemma and the fact that $(\varepsilon_i, Z_i)$ are i.i.d., we obtain that, for any $c \in \mathbb{R}$,

$$\begin{aligned}
\mathbb{E}\left[\exp\left(c(X_\varphi - X_{\varphi'})\right)\right] &= \mathbb{E}\left[\exp\left(\frac{c}{\sqrt{n}} \sum_{i=1}^{n} \varepsilon_i\left(\varphi(Z_i) - \varphi'(Z_i)\right)\right)\right] \\
&= \left\{\mathbb{E}\left[\exp\left(\frac{c}{\sqrt{n}}\varepsilon_1\left(\varphi(Z_1) - \varphi'(Z_1)\right)\right)\right]\right\}^n \\
&\leq \exp\left(\frac{c^2 d_\Phi(\varphi, \varphi')^2}{2}\right).
\end{aligned}$$

Hence, $X$ is subgaussian with respect to $d_\Phi(\varphi, \varphi')$, and therefore the Rademacher average $R_n(\Phi)$ can be upper-bounded by the Dudley entropy integral (see e.g., Talagrand, 2021):

$$R_n(\Phi) \leq \frac{12}{\sqrt{n}} \int_0^\infty \sqrt{\log \mathcal{N}(\Phi, d_\Phi, u)} \, du,$$

where $\mathcal{N}(\Phi, d_\Phi, \cdot)$ denotes the covering numbers of $(\Phi, d_\Phi)$.

From the definition of $d_\Phi(\varphi, \varphi')$ above, we have

$$\mathcal{N}(\Phi, d_\Phi, u) \leq \mathcal{N}\left(\mathcal{F}, \|\cdot\|_{L_\infty(d_0)}, \frac{u}{2}\right) \cdot \mathcal{N}\left([0, \, L_\ell L_f R^{1-p}], |\cdot|, \frac{u}{2\,\mathscr{D}_{\mathcal{Z}}^p}\right),$$

and therefore

$$R_n(\Phi) \leq \frac{12}{\sqrt{n}} \left\{ \int_0^\infty \sqrt{\log \mathcal{N}\left(\mathcal{F}, \|\cdot\|_{L_\infty(d_0)}, \frac{u}{2}\right)}\, du + \int_0^\infty \sqrt{\log \mathcal{N}\left([0, \, L_\ell L_f R^{1-p}], |\cdot|, \frac{u}{2\,\mathscr{D}_{\mathcal{Z}}^p}\right)}\, du \right\}.$$

Since $[0, \, L_\ell L_f R^{1-p}]$ is a compact interval, it is straightforward to upper-bound the second integral:

$$\int_0^\infty \sqrt{\log \mathcal{N}\left([0, \, L_\ell L_f R^{1-p}], |\cdot|, \frac{u}{2\,\mathscr{D}_{\mathcal{Z}}^p}\right)}\, du \leq 2\, L_\ell L_f R^{1-p}\, \mathscr{D}_{\mathcal{Z}}^p \int_0^{1/2} \sqrt{\log(1/u)}\, du \tag{29}$$

$$= 2c\, L_\ell L_f R^{1-p}\, \mathscr{D}_{\mathcal{Z}}^p,$$

where $L_\ell L_f R^{1-p}$ is the length of the interval $[0, \, L_\ell L_f R^{1-p}]$, and

$$c = \frac{1}{2}\sqrt{\log 2} + \sqrt{\pi}\, \mathrm{erfc}\!\left(\sqrt{\log 2}\right) < 1.$$

Thus,

$$R_n(\Phi) \leq \frac{12}{\sqrt{n}} \left\{ \int_0^\infty \sqrt{\log \mathcal{N}\left(\mathcal{F}, \|\cdot\|_{L_\infty(d_0)}, \frac{u}{2}\right)}\, du + 2\, L_\ell L_f R^{1-p}\, \mathscr{D}_{\mathcal{Z}}^p \right\}.$$

Using Definition 3.6 of $\mathfrak{C}_{\infty(d_0)}(\mathcal{F}) := \int_0^\infty \sqrt{\log \mathcal{N}\left(\mathcal{F}, \|\cdot\|_{L_\infty(d_0)}, \varepsilon\right)}\, d\varepsilon$ and integration by substitution (reverse chain rule), we obtain

$$R_n(\Phi) \leq \frac{24}{\sqrt{n}} \left( \mathfrak{C}_{\infty(d_0)}(\mathcal{F}) + L_\ell L_f R^{1-p}\, \mathscr{D}_{\mathcal{Z}}^p \right).$$

Plugging this into Equation 28 yields

$$\mathscr{R}(\mathrm{Tr}(d_0, \widehat{\theta}_T), \widehat{\theta}_T) - \mathscr{R}(\mathrm{Tr}(\widehat{d}_0, \widehat{\theta}_T), \widehat{\theta}_T) \leq \frac{48}{\sqrt{n}} \left( \mathfrak{C}_{\infty(d_0)}(\mathcal{F}) + L_\ell L_f R^{1-p}\, \mathscr{D}_{\mathcal{Z}}^p \right) + F\sqrt{\frac{2\log(2/\delta)}{n}} + 2L_\ell R,$$

with probability $1 - 3\delta$ via union bound, as required. $\qquad\square$

### D.8. Proof of Theorem 3.15

We restate the theorem for readability.

**Theorem** (Generalization Gap II, **RQ2**). *Under Conditions 3.1–3.3 and 3.14, we obtain the same bound on* $\mathscr{R}(\mathrm{Tr}(d_0, \widehat{\theta}_T), \widehat{\theta}_T) - \mathscr{R}(\mathrm{Tr}(\widehat{d}_0, \widehat{\theta}_T), \widehat{\theta}_T)$ *as in Theorem 3.13 but with* $B2^{p-1}(1 + \mathscr{D}_{\mathcal{Z}}/R)^p$ *instead of* $L_\ell L_f R^{1-p}$.

*Proof.* The reasoning mirrors the proof of Theorem 3.13, except that we upper bound $\widehat{\lambda}$ by $B2^{p-1}(1 + \mathscr{D}_{\mathcal{Z}}/R)^p$, where $B$ is from Condition 3.14 and

$$R := \max\left\{ \left(\frac{m}{n} + q(\delta)\sqrt{\frac{\frac{m}{n}(1 - \frac{m}{n})}{n}}\right)^{\frac{1}{p}} \mathscr{D}_{\mathcal{Z}}, \; \frac{\varepsilon^T(1 + L_a)^T - 1}{\varepsilon(1 + L_a) - 1}\left(\frac{m}{n}\right)^{\frac{1}{p}} \mathscr{D}_{\mathcal{Z}} \right\},$$

as in the proof of Theorem 3.13.

Recall $\widehat{\lambda} := \underset{\lambda}{\arg\min}\left\{ \lambda R^p + \mathbb{E}_{\widehat{d}_0}\left[\varphi_{\lambda, \widehat{\theta}_T}(Z)\right] \right\}$. Since $\varphi_{\lambda, \widehat{\theta}_T} \geq 0$ for all, we clearly have $\widehat{\lambda} \leq \sup_{W_p(d_0, d) \leq R} \mathscr{R}(d, \theta)/R^p$.

Further recall from Equation 25, the dual characterization of a locally supremum risk by Gao & Kleywegt (2023):

$$\sup_{W_p(d_0, d) \leq R} \mathscr{R}(d, \theta) = \min_{\lambda \geq 0}\left\{ \lambda R^p + \mathbb{E}_{d_c}\left[\varphi_{\lambda, \theta}(Z)\right] \right\},$$

where

$$\varphi_{\lambda,\theta}(z) = \sup_{z' \in \mathcal{Z}} \left\{ \ell(f_\theta(x'), y') - \lambda d_{\mathcal{Z}}^p(z, z') \right\}.$$

Thus,

$$\inf_\theta \sup_{W_p(d_0, d) \leq R} \mathscr{R}(d, \theta) = \inf_\theta \min_{\lambda \geq 0} \left\{ \lambda R^p + \mathbb{E}_{d_c} \left[ \varphi_{\lambda,\theta}(Z) \right] \right\}.$$

Hence, we can apply Lemma 2 of Lee & Raginsky (2018) to yield $\widehat{\lambda} \leq B 2^{p-1} (1 + \mathscr{D}_{\mathcal{Z}}/R)^p)$. We can the define the function class

$$\tilde{\Phi} := \{ \varphi_{\lambda,\theta} : \lambda \leq B 2^{p-1} (1 + \mathscr{D}_{\mathcal{Z}}/R)^p), f_\theta \in \mathcal{F} \},$$

instead of

$$\Phi := \{ \varphi_{\lambda,\theta} : \lambda \leq L_\ell L_f R^{1-p}, f_\theta \in \mathcal{F} \},$$

as in the proof of Theorem 3.13. The remainder of the proof directly follows from applying the reasoning in the proof of Theorem 3.13 to $\tilde{\Phi}$ in place of $\Phi$.

In particular, in place of Equation 29, we get

$$\int_0^\infty \sqrt{\log \mathcal{N}\left( [0, B 2^{p-1}(1 + \mathscr{D}_{\mathcal{Z}}/R)^p)], |\cdot|, \frac{u}{2 \mathscr{D}_{\mathcal{Z}}^p} \right)} \, du \leq 2 B 2^{p-1}(1 + \mathscr{D}_{\mathcal{Z}}/R)^p) \mathscr{D}_{\mathcal{Z}}^p \int_0^{1/2} \sqrt{\log(1/u)} \, du$$

$$= 2c \, B 2^{p-1}(1 + \mathscr{D}_{\mathcal{Z}}/R)^p) \mathscr{D}_{\mathcal{Z}}^p,$$

where $B 2^{p-1}(1 + \mathscr{D}_{\mathcal{Z}}/R)^p)$ is the length of the interval $[0, B 2^{p-1}(1 + \mathscr{D}_{\mathcal{Z}}/R)^p)]$, and

$$c = \frac{1}{2}\sqrt{\log 2} + \sqrt{\pi} \, \mathrm{erfc}\left( \sqrt{\log 2} \right) < 1.$$

Thus,

$$R_n(\tilde{\Phi}) \leq \frac{12}{\sqrt{n}} \left\{ \int_0^\infty \sqrt{\log \mathcal{N}\left( \mathcal{F}, \|\cdot\|_{L_\infty(d_0)}, \frac{u}{2} \right)} \, du + 2 B 2^{p-1}(1 + \mathscr{D}_{\mathcal{Z}}/R)^p) \mathscr{D}_{\mathcal{Z}}^p \right\}.$$

Using Definition 3.6 of $\mathfrak{C}_{\infty(d_0)}(\mathcal{F}) := \int_0^\infty \sqrt{\log \mathcal{N}\left( \mathcal{F}, \|\cdot\|_{L_\infty(d_0)}, \varepsilon \right)} \, \mathrm{d}\varepsilon$ and integration by substitution, we obtain

$$R_n(\tilde{\Phi}) \leq \frac{24}{\sqrt{n}} \left( \mathfrak{C}_{\infty(d)}(\mathcal{F}) + B 2^{p-1}(1 + \mathscr{D}_{\mathcal{Z}}/R)^p) \mathscr{D}_{\mathcal{Z}}^p \right).$$

Analogous to the proof of Theorem 3.13, we then get

$$\mathscr{R}(\mathrm{Tr}(d_0, \widehat{\theta}_T), \widehat{\theta}_T) - \mathscr{R}(\mathrm{Tr}(\widehat{d}_0, \widehat{\theta}_T), \widehat{\theta}_T) \leq \frac{48}{\sqrt{n}} \left( \mathfrak{C}_{\infty(d)}(\mathcal{F}) + B 2^{p-1}(1 + \mathscr{D}_{\mathcal{Z}}/R)^p) \mathscr{D}_{\mathcal{Z}}^p \right) + F \sqrt{\frac{2 \log(2/\delta)}{n}} + 2 L_\ell R,$$

with probability $1 - \delta$, as required.

$\square$

### D.9. Proof of Theorem 3.16

For ease of exposition, we start by restating the result.

**Theorem** (Cumulative Performative Excess Risk Bound). *Under Conditions 3.1–3.3, we have that,*

$$\sum_{t=T}^{\tilde{T}} \left( \mathscr{R}(d_t, \theta_t) - \inf_{\theta \in \Theta} \mathscr{R}(d_t, \theta) \right) \leq \mathscr{B}(T, m, n) + (\tilde{T} - T + 1) L_\ell L_a \left( \frac{m}{n} + q(\delta) \sqrt{\frac{\frac{m}{n}(1 - \frac{m}{n})}{n}} \right)^{\frac{1}{p}} \mathscr{D}_{\mathcal{Z}},$$

*for any $\tilde{T} \geq T$, with probability $1 - 5\delta$, where $d_t = \mathrm{Tr}(d_{t-1}, \theta_t)$ for $t \geq T$, $\theta_T = \widehat{\theta}_T$, $d_{T-1} = d_0$ and $\mathscr{B}(T, m, n)$ is the performative excess risk bound from Theorem 3.10.*

*Proof.* The idea of the proof is to leverage the recursive argument that $\theta_t$ is the (population) risk minimizer under $d_{t-1}$, which determines $d_t$ via the unknown Tr.

We want to bound the *cumulative performative excess risk*

$$\sum_{t=T}^{\tilde{T}} \mathscr{R}(d_t, \theta_t) - \inf_{\theta \in \Theta} \mathscr{R}(d_t, \theta),$$

with $d_t = \text{Tr}(d_{t-1}, \theta_t)$, $\theta_T = \widehat{\theta}_T$ and $d_{T-1} = d_0$.

This term is equivalent to

$$\left( \mathscr{R}(\text{Tr}(d_0, \widehat{\theta}_T), \widehat{\theta}_T) - \inf_{\theta \in \Theta} \mathscr{R}(\text{Tr}(d_0, \widehat{\theta}_T), \theta) \right) + \sum_{t=T+1}^{\tilde{T}} \mathscr{R}(d_t, \theta_t) - \inf_{\theta \in \Theta} \mathscr{R}(d_t, \theta).$$

We have an upper bound with high probability on $\mathscr{R}(\text{Tr}(d_0, \widehat{\theta}_T), \widehat{\theta}_T) - \inf_{\theta \in \Theta} \mathscr{R}(\text{Tr}(d_0, \widehat{\theta}_T), \theta)$ by Theorem 3.10. In order to bound $\sum_{t=T+1}^{\tilde{T}} \mathscr{R}(d_t, \theta_t) - \inf_{\theta \in \Theta} \mathscr{R}(d_t, \theta)$, we first observe from Definition 2.3 that $\theta_t$ is the (population) risk minimizer on $d_{t-1}$. In other words,

$$\theta_t \in \arg \inf_{\theta \in \Theta} \mathscr{R}(d_{t-1}, \theta),$$

and analogously

$$\theta_{t+1} \in \arg \inf_{\theta \in \Theta} \mathscr{R}(d_t, \theta).$$

We can thus write $\sum_{t=T+1}^{\tilde{T}} \mathscr{R}(d_t, \theta_t) - \inf_{\theta \in \Theta} \mathscr{R}(d_t, \theta) = \sum_{t=T+1}^{\tilde{T}} \mathscr{R}(d_t, \arg \inf_{\theta \in \Theta} \mathscr{R}(d_{t-1}, \theta)) - \inf_{\theta \in \Theta} \mathscr{R}(d_t, \theta)$, or more helpfully,

$$\sum_{t=T+1}^{\tilde{T}} \mathscr{R}(d_t, \theta_t) - \inf_{\theta \in \Theta} \mathscr{R}(d_t, \theta) = \sum_{t=T+1}^{\tilde{T}} \mathscr{R}(d_t, \theta_t) - \mathscr{R}(d_t, \theta_{t+1}).$$

Since Tr modifies a share $s \in [0, 1]$ of the population and $\mathcal{Z}$ is bounded, an argument analogous to the in-sample case gives

$$\|\theta_t - \theta_{t+1}\| \leq L_a \, s^{1/p} \mathscr{D}_{\mathcal{Z}},$$

where $L_a$ is the Lipschitz constant of the map $P \mapsto \arg \min_{\theta \in \Theta} \mathscr{R}(P, \theta)$. We need to estimate the performative propensity in the population, i.e., the share $s$ of the population reacting to predictions. Lemma 3.9 gives

$$s < \frac{m}{n} + q(\delta) \sqrt{\frac{m}{n^2} (1 - \frac{m}{n})},$$

with probability $1 - \delta$, where $q(\delta) = q_{1-\delta}$ is the $(1 - \delta)$-quantile of a standard normal distribution and $m/n$ is the share of the sample that changes. Using this high-probability upper bound on $s$ as above, we obtain

$$\|\theta_t - \theta_{t+1}\| \leq L_a \left( \frac{m}{n} + q(\delta) \sqrt{\frac{\frac{m}{n}(1 - \frac{m}{n})}{n}} \right)^{\frac{1}{p}} \mathscr{D}_{\mathcal{Z}},$$

with probability at least $1 - \delta$. Since $\ell$ is Lipschitz in $\theta$ (and thus $\mathscr{R}$ as an integral over $\ell$), we have

$$\mathscr{R}(d_t, \theta_t) - \mathscr{R}(d_t, \theta_{t+1}) \leq L_\ell \|\theta_t - \theta_{t+1}\|,$$

and hence

$$\mathscr{R}(d_t, \theta_t) - \mathscr{R}(d_t, \theta_{t+1}) \leq L_\ell L_a \left( \frac{m}{n} + q(\delta) \sqrt{\frac{\frac{m}{n}(1 - \frac{m}{n})}{n}} \right)^{\frac{1}{p}} \mathscr{D}_{\mathcal{Z}},$$

with probability at least $1 - \delta$. This implies

$$\sum_{t=T-1}^{\tilde{T}} \mathscr{R}(d_t, \theta_t) - \mathscr{R}(d_t, \theta_{t+1}) \leq (\tilde{T} - T + 1) L_\ell L_a \left( \frac{m}{n} + q(\delta)\sqrt{\frac{\frac{m}{n}(1 - \frac{m}{n})}{n}} \right)^{\frac{1}{p}} \mathscr{D}_{\mathcal{Z}},$$

with probability at least $1 - \delta$.

With the bound on $\mathscr{R}(\mathrm{Tr}(d_0, \widehat{\theta}_T), \widehat{\theta}_T) - \inf_{\theta \in \Theta} \mathscr{R}(\mathrm{Tr}(d_0, \widehat{\theta}_T), \theta)$, Theorem 3.10 implies that

$$\sum_{t=T}^{\tilde{T}} \mathscr{R}(d_t, \theta_t) - \inf_{\theta \in \Theta} \mathscr{R}(d_t, \theta) \leq \mathscr{B}(T, m, n) + (\tilde{T} - T + 1) L_\ell L_a \left( \frac{m}{n} + q(\delta)\sqrt{\frac{\frac{m}{n}(1 - \frac{m}{n})}{n}} \right)^{\frac{1}{p}} \mathscr{D}_{\mathcal{Z}},$$

with probability at least $1 - 5\delta$, where $\mathscr{B}(T, m, n)$ is the performative excess risk bound from Theorem 3.10. $\qquad \square$

