# OpenReview forum: "Performative Learning Theory"
_ICML.cc/2026/Conference — ICML 2026 regular_

### Official Review · Reviewer_ThZS · 2026-03-09

**Soundness:** 4
**Presentation:** 3
**Significance:** 4
**Originality:** 4
**Overall Recommendation:** 5
**Confidence:** 4

**Summary:**

This paper presents a systematic study on the learnability with performativity. Starting by two running examples, the authors first introduce several major settings with different types of performative effects and sample regimes. The authors then utilize statistical tools to analyze generalization error and excess risk bounds under these settings. The authors discuss implications regarding theoretical results, including the tendency of convergence in different regimes, the fundamental trade-off between exploring more samples and achieving better generalization error bound, and the usage of performative samples beyond just searching in the hypothesis space. The authors also illustrate the generalization bounds they have derived on a dataset.

**Compliance With Llm Reviewing Policy:**

Affirmed.

**Final Justification:**

After reading the response to me as well as to the other reviewers, my concerns have been adequately addressed, and I decided to keep my score.

**Key Questions For Authors:**

- (Q1) In Section 3.2 just after Theorem 3.10, the authors say that "retraining on $\hat{d}_0$, \cdots, $\hat{d}_T$ can still improve generalization bounds". I don't understand how this claim is summarized from previous theorems. Can the authors provide a more detailed explanation?
- (Q2) I can get the difference between performative learnability and previous studies on performative prediction. However, I am interested in the possibilities that the analysis on performative prediction (specifically, those regarding non-population bounds such as the optimization error or regret bounds over a sequence of samples) and the analysis in this paper can be composed to form a complete analysis of the process "learning in a performative environment", which incorporates not only the statistical properties that affect the complexity of learning which are mentioned in this paper, but also the properties of the learning / optimization algorithm itself that probably provide a sharper convergence rate over sequence of samples. Could the authors provide a brief discussion on this point?

**Limitations:**

yes

**Strengths And Weaknesses:**

- Strengths: The paper is technically sound and is clearly written. Specifically, the categorization of different research problems provide a clear understanding on the problem of learnability in the presence of performativity. The paper addresses a definitely important problem, that is, how to formulate and analyze the "endogeneous effect" of the environment in a general and systematic manner? Besides RL and performative prediction (decision-dependent optimization), this paper proposes a learnability framework that touches some new aspects of this problem.
- Weaknesses: I believe this is a good paper and is worth acceptance. Below are just several suggestions.
  - (W1) The authors mentioned in abstract that in the analysis, the self-negating and self-fulfilling phenomena are cast into the min-max and min-min risk functionals in Wasserstein space respectively, while in main text such a casting technique is only briefly mentioned. If the authors feel that the casting technique is worth mentioning as a technical contribution, then it would be better if it can be summarized into a technical lemma and formally presented in main text.
  - (W2) Section 4 seems less informative, i.e., the illustration of generalization error bounds which are explicit mathematical expressions is straightforward. It would be better if the authors can provide the comparison between the true generalization error on some problems and the generalization error bounds derived in the paper.

   I've also found a possible typo: just before Definition 2.3, on the right-hand-side in the definition of the stateful performative risk of $\theta$, is it $\mathbb{E}_{Z\sim\texttt{Tr}(\theta, d_{t-1})$ rather than $\mathbb{E}_{Z\sim\texttt{Tr}(\theta_t, d_{t-1})$?

---

> ### Author Rebuttal · Authors · 2026-03-27
>
> Thank you very much for the thoughtful and constructive assessment of our work. We are encouraged by your view that the paper presents “a systematic study on learnability with performativity” and provides “a clear understanding” thereof. We are grateful you found the work “technically sound”, “clearly written” and addressing “a definitely important problem.”
>
> $~$
>
> Re **(W1)** We fully agree this novel technique deserves a more prominent treatment in the main text. We add a dedicated lemma and point to the underlying proof mechanism: Beginning with Eq. (78), the performative generalization gap is rewritten as a local min-max / min-min functional in Wasserstein space. This is different from standard Wasserstein distributionally robust optimization because it couples a *worst-case population* risk with a *best-case sample* risk. The dual reformulation in Equations (80)–(83) and the subsequent symmetrization / Rademacher step are therefore essential and non-standard, see our reply to reviewer Ehak.
>
> Re **(W2)** We 100% agree that the role of Section 4 needed to be stated more clearly. We now make explicit that its purpose is not to claim tightness in applications, but to illustrate how the different terms in the bound behave and how the *change–learn trade-off* we discovered manifests in practice. At the same time, we strengthened the discussion by also reporting the observed test error in our semi-simulation study. We agree this makes the illustration section more informative.
>
> Re **(Q1)** Great question! We make this claim more explicit in Corollary 3.11 right below the mentioned paragraph. Intuition: If you deploy a model on a performatively changing sample $T$ times, you *observe* $T$ performative shifts of the sample. This allows you to better estimate (by reducing the variance of the estimator) the performative shift you will suffer when you deploy the model out-of-sample. (It increases the “sample size” of that estimator.) Since the high probability bounds depend on the tail behavior of that estimator’s distribution, obviously a smaller variance for the estimator will shrink the bound. We explain this more thoroughly in the revision. Many thanks!
>
>
> Re **(Q2)** We also appreciate your suggestion to say more about how our analysis interfaces with existing work on optimization error, regret and performative prediction. We now do this more concretely. Our analysis is indeed complementary to optimization- or regret-based analyses such as Perdomo et al. (2020) and Brown et al. (2022). If one additionally has an optimization guarantee for approximate ERM/RERM, or a regret bound for a performative learner, then the overall bounds simply decompose into three components: the classical statistical term (see discussion below Thm. 3.7) + a performative shift term (ibid.) + a optimization/regret term. So rather than competing with that line of work, our results provide the statistical component needed for a comprehensive theory of learning in performative environments. We believe this clarification improves the presentation of our results substantially. Since it arose directly from your comment, we’d like to thank you wholeheartedly.
>
> Finally, thank you for pointing out the **typo right before Definition 2.3**. We have corrected the RHS as suggested so that model and induced distribution are indexed consistently.
>
> $~$
>
> **To sum up**, we thank you once more for the positive and constructive assessment. We especially appreciate that you evaluated the paper as “excellent” in soundness, significance, and originality, while the presentation was where you saw room for improvement (albeit being “clearly written” generally). We took that assessment very seriously and have endeavored to improve the presentation, as we outlined above.
>
> **Your suggestions directly helped** improve in that regard: the revised manuscript now states the min-max/min-min perspective in Wasserstein space more explicitly; clarifies the interpretation of Corollary 3.11 (“Improving Bounds Under Performativity”) and the discussion after Theorem 3.10; explains the relation to optimization/regret analyses; and reports empirical test errors in our illustrative case study.
>
> Since these clarifications **have now been incorporated** directly and concretely, we hope the revised manuscript better reflects the strength that you already saw in the underlying contribution. *In light of these improvements, we would be very grateful if you could consider whether the paper is now closer to a strong accept than to an accept.*
>
> We would, of course, be happy to address any remaining questions you may have.

---

> > ### Author Rebuttal · Reviewer_ThZS · 2026-04-01
> >
> > Thanks the authors for their response. After reading the response to me as well as to the other reviewers, my concerns have been adequately addressed, and I decided to keep my score.

---

> > > ### Author Response · Authors · 2026-04-03
> > >
> > > Thank you very much for getting back to us!
> > >
> > > We are glad that our response to you (and to other reviewers) addressed your concerns adequately. If there is still any remaining concern or room for improvement, we would of course be very grateful for a brief indication and would be happy to work towards further improving our paper in that direction. Thanks in advance!
> > >
> > > All in all, we'd like to thank you once more for helping us improve this work. We are very grateful for your thoughtful reading and constructive feedback.

---

### Official Review · Reviewer_Ehak · 2026-03-10

**Soundness:** 3
**Presentation:** 3
**Significance:** 2
**Originality:** 2
**Overall Recommendation:** 4
**Confidence:** 3

**Summary:**

This paper studies generalization in performative learning, where the data distribution can change over time and the deployed model itself may contribute to that change. The paper aims to formulate “generalizability” in such performative settings rigorously. It introduces four benchmark metrics to quantify different notions of generalization under performativity. Under a set of assumptions, including strong convexity of the loss, Lipschitz Wasserstein control on distribution drift, and smoothness conditions on the model class, the paper derives upper bounds for these benchmark metrics. The paper also provides an empirical illustration on real-world datasets.

**Compliance With Llm Reviewing Policy:**

Affirmed.

**Final Justification:**

Same as my Acknowledgement.

**Key Questions For Authors:**

See weaknesses section.

**Strengths And Weaknesses:**

**Strengths:**

1. The main conceptual contribution of the paper appears to be the formulation of the benchmark metrics RQ1–RQ4, which may inspire follow-up work in this area.
2. The paper provides explicit non-asymptotic upper bounds on these generalization metrics, which is non-trivial, even though I did not verify the full proofs due to time constraints.

**Weaknesses:**

1. It seems to me that essentially all of the upper bounds ultimately rely on the Wasserstein control assumption in Condition 2.3. I am not convinced that this assumption is as “minimal” or as natural as the paper suggests. At the very least, the paper should provide concrete settings in which this condition holds naturally, or show how it follows from more interpretable structural assumptions.
2. Although the paper provides a set of non-trivial upper bounds, many of them seem to follow from combining fairly standard Wasserstein-based arguments. More importantly, the authors do not justify the tightness of these bounds through matching lower bounds or sharpness results. As a result, it remains unclear how informative the bounds are in concrete models, or even whether they vanish in meaningful regimes.
3. The empirical section does not really validate the theory. It mainly serves as an illustration of the proposed bounds on a real-world-inspired dataset, rather than demonstrating that the bounds are tight, practically informative, or predictive of actual generalization behavior.

**Overall:**
Although the paper clarifies several generalization regimes and provides some interesting bounds for performative learning, I am not convinced that it is a particularly strong theory paper, nor that it delivers a deep conceptual leap. My recommendation is *weak reject*, though it is close to borderline.

---

> ### Author Rebuttal · Authors · 2026-03-27
>
> Thanks for engaging with our work! We appreciate the encouraging feedback, esp. on the conceptualization of learning theory under performativity via RQ1-4 and the non-triviality of our results.
>
> $~$
>
> We fully agree the previous description of **Wasserstein sensitivity** (Cond. 3.2, which we assume is meant here) as “minimal” is misleading. We meant “general”/“generic,” in the sense that this assumption is used throughout basic PP literature. See Def. 3.1 in Perdomo et al. (2020), eq. (A1) in Miller et al. (2021), Def. 1 in Brown et al. (2022), the even more restrictive (A1) in Mofakhami et al. (2023), and Assumption 3.1(d) in Li et al. (2025b) (see refs. in paper), as well as Def. 1 in [1], Assumption 1.2 in [2], Assumption A3 in [3], Assumption 2 in [4], Assumption 2.2 in [5], Condition W1 in [6], Def. 2.3 in [7], and more… (truncated due to char. limit)
>
> These works use Cond. 3.2 to cover a wide range of transition maps $\mathrm{Tr}$, while works like Mendler-Dunner et al. (2023) and Kirev et al. (2025) consider specific, even fully parametric, explicit functional forms of $\mathrm{Tr}$. Hence we (misleadingly) called Cond. 3.2 “minimal.” In revision, we replace “minimal” by “the generic setup by Perdomo et al. (2020), Brown et al. (2022), Li et al. (2025b) and [1-7].” We also refer to the discussion right below Eq. 1 in [8], where it is said that Wasserstein sensitivity’s “generality enables us to write a broad range of prediction problems—including supervised learning, strategic classification, and causal inference—as special cases of performative prediction” (including further refs.).
>
> While Cond. 3.2 is standard in PP, we still 100% agree that **interpretable sufficient conditions / examples for Cond. 3.2 will improve presentation**. In line with Sec. 1.1, we do this for our two running examples and also point to further generic examples where Cond. 3.2 holds, such as Remark 3.2 in Perdomo et al. (2020). Due to char. limits, we show only one spelled-out instance here, namely running example (A) of routing apps: if the next-round distribution is generated by a Markov kernel $K_\theta(\cdot\mid z)$ that is Lipschitz in both $z$ and $\theta$, then $\mathrm{Tr}$ satisfies joint Wasserstein sensitivity. This is one response studied in routing settings where recommendation changes induce bounded changes in traffic behavior (Cabannes, 2019; Benenati and Grammatico, 2024).
>
> Moreover, we respectfully **disagree that the results are obtained by merely combining standard Wasserstein-based arguments**. Eq. (78) shows that, unlike in standard distributionally robust optimization (DRO), our inf-sup/inf-inf risk functionals in Wasserstein space are between *population supremum* risk and *sample infimum* risk, so a direct Kantorovich-Rubinstein or a standard DRO argument is NOT enough. We need original empirical process theory for these dual functionals (see, e.g., symmetrization/Rademacher steps for the $\Phi$-indexed process in lines 1756–1807 and Eqs. (82)–(88)).
>
> We firmly believe **matching lower bounds / sharpness** results, while interesting and worth pursuing, are beyond the paper’s scope. The main focus of this work is to embed the whole PP literature into learning theory, which is itself a substantial endeavour (see Sec. 2.1), and answer resulting research questions by non-trivial gen. bounds (as you acknowledge).
>
> In our revision, we emphasize more clearly that **our bounds *are* informative** in various regimes. For example, in Thm. 3.10, the ordinary statistical terms vanish with $n$, while the perf. contribution scales with the perf. fraction $m/n$ and the accumulated distortion factor. Hence, if $m=o(n)$ and $\varepsilon\kappa/\gamma<1$, the bound vanishes; if $m/n$ is held fixed, the extra term quantifies the explicit price of performativity relative to standard generalization.
>
> $~$
>
> We again **thank you for helping improve our paper**; we believe it really did. Given that you positioned our paper “close to borderline,” we’d be grateful if you’d re-evaluate your assessment in light of our clarifications (esp. w.r.t. Cond. 3.2) and suggested changes. We would, of course, be happy to address any remaining questions you may have. Thank you!
>
> $~$
>
> Addit. Refs:
>
> [1] Mendler-Dunner et al. "Stochastic optimization for performative prediction." NeurIPS 2020
>
> [2] Jagadeesan et al., “Regret Minimization with Performative Feedback.” ICML 2022
>
> [3] Li et al., “Multi-agent Performative Prediction with Greedy Projection and Consensus Seeking” NeurIPS 2022
>
> [4] Narang et al., “Multiplayer Performative Prediction: Learning in Decision-Dependent Games.” JMLR 2023
>
> [5] Wang et al., “Network Effects in Performative Prediction Games” ICML 2023
>
> [6] Li and Wai, “Stochastic Optimization Schemes for Performative Prediction with Nonconvex Loss.” NeurIPS 2024
>
> [7] Kun et al. "Addressing polarization and unfairness in performative prediction." AAAI 2026
>
> [8] Kim et al. "Making decisions under outcome performativity.” ITCS 2023

---

> > ### Author Rebuttal · Reviewer_Ehak · 2026-04-01
> >
> > I thank the authors for their responses. I believe the main weakness of the paper is the interpretability of the bounds, a concern that is also shared by Reviewer rjij. In particular, it is difficult to assess the tightness (or even the usefulness) of upper bounds in the absence of any sharpness analysis, even in simple toy models. Without such analysis, it is hard to tell whether the bounds merely follow from the assumptions or whether they actually capture meaningful structure of the problem.
> >
> > That said, given the authors’ promise to add running examples to better illustrate the informativeness of the bounds, I am willing to adjust my score to 4. However, I still do not view this as a strong theory paper.

---

> > > ### Author Response · Authors · 2026-04-03
> > >
> > > Thank you very much for your thoughtful follow-up and for reconsidering our paper in light of the rebuttal. We greatly appreciate your careful engagement with our clarifications and your willingness to adjust your score.
> > >
> > > We also appreciate your frank assessment of the remaining limitation, which we take very seriously. We fully agree matching lower bounds / sharpness results are important, but require significant novel work, and as such are an excellent avenue for future research.
> > >
> > > As promised, we add interpretable sufficient conditions / examples for Cond. 3. We also extend the two running examples substantially and give additional examples to illustrate explictly that our bounds are informative, see above. We are glad you see this as steps towards better interpretability and illustration of our bounds.
> > >
> > > Thank you again for the constructive feedback. Your comments have helped us a lot to improve the presentation of our work.

---

### Official Review · Reviewer_rjij · 2026-03-11

**Soundness:** 3
**Presentation:** 2
**Significance:** 3
**Originality:** 2
**Overall Recommendation:** 4
**Confidence:** 3

**Summary:**

This is a theory paper about performative learning.  Performative predictions influence the outcome they aim to forecast. For example, drivers are known to avoid routes that have predicted congestion, thereby rendering these predictions less accurate. Apart from self-defeating examples like the one with routing that was just mentioned, changes can go in the other direction as well, making them self-fulfilling.  An example in this latter direction is given along the lines of job-finding.  For example, in a job center where one predicts that certain individuals may have long unemployment times, it can be suggested to them to perform some job training, which in turn will increase their probability of finding a job.  In this direction the authors study repeated empirical risk minimization (RERM), where learning is now seen in form of rounds and in each round the model is learning from examples that are coming from a "drifted" distribution due to the performative effect.  The authors study four research questions involving the classical and performative excess risk, as well as how models trained on specific samples provide generalization guarantees over the whole population, by potentially learning a new model in different phases of the learning process.  The authors provide guarantees in several cases under the assumption that the loss function is strongly convex, the performative shift in the distribution is not large (bounded by the Wasserstein distance), and the loss function has some additional nice properties (Lipschitzness). The authors also provide an illustration of their bounds in the job-seeking example.

**Compliance With Llm Reviewing Policy:**

Affirmed.

**Final Justification:**

I have increased my recommendation from weak reject (3) to weak accept (4) given that the authors will provide the additional clarifications and experiments in the final version of their paper.  Please see my acknowledgement as well.

**Key Questions For Authors:**

**Q1.**  Can you translate any of the generalization results that you have in terms of more "traditional" generalization results that connect number of examples and excess (performative) risk?  Also, relatedly, does it matter if one performs proper learning, learning in the realizable case, or otherwise?

**Limitations:**

I think the main limitation of the paper is that all the generalization bounds that the authors provide involve terms that need significant lookup and back-and-forth in the paper and even then I find them hard to understand.  For example, none of the results has the form of standard PAC bounds that relate the number of examples with the excess (performative) risk.  In addition, it is not entirely clear to me what is permissible and what is not between different rounds of learning when we learn from samples from a shifted distribution as a result of learning a model based on a sample from a previous distribution.  Perhaps the notation is heavy and not well-explained in the preliminaries section.

**Strengths And Weaknesses:**

The paper appears to be sound.  To the extent that I checked the ideas and the proofs, they appear to be correct.  However, I believe that the presentation can be improved.  While I appreciate the fact that the authors have separated the four research questions that they want to answer and have also provided Table 1 to explain different cases, nevertheless, the notation, terminology, and the bounds appear to be highly non-standard.  For example, we do not see some relationship between the sample size and the excess generalization error $\epsilon$ that would be a pretty much standard relationship for generalization bounds.  Of course, there is merit in this work nevertheless as this is an interesting real-world problem.  As for the originality of the approach, I believe it is fair as well.

---

> ### Author Rebuttal · Authors · 2026-03-27
>
> Thank you for your thoughtful comments. We appreciate that, despite your reservations about *presentation*, you found the paper fairly original, technically “sound,” and that the proofs “appear to be correct.” We also value your recognition that this is “an interesting real-world problem” and that “there is merit in this work.” In the camera-ready version, we improve the presentation to make the paper more accessible to a broader audience.
>
> $~$
>
> The main reservation seems to be **how our generalization bounds relate to “standard PAC bounds that map the number of examples ($n$) to the excess risk.”** In fact, our excess risk bounds *do* shrink with $n$, much like standard PAC bounds. All terms in our performative excess risk bounds (Thms. 3.7 and 3.10) explicitly depend on sample size $n$. We discuss this right below Thm. 3.7 and note below Thm. 3.10 that the same dependence carries over there. Specifically, for fixed $m$ (the number of units in the sample that performatively react to predictions; see Lemma 3.5), the ordinary statistical terms vanish with $n$, and the remaining terms quantify the explicit *cost of performativity* relative to standard PAC settings. If $m=o(n)$ and $\varepsilon\kappa/\gamma<1$, then the entire bound vanishes asymptotically, including the performative part.
>
> What might have caused this misunderstanding is that we did not explain the difference between $n$ and $m$ clearly enough in this particular section. In the revision, we will do this more pominently right below both Thms. 3.7 and 3.10 and add citations to standard PAC bounds in the discussion below Thm. 3.7.
>
> Your question about **proper / realizable learning** is also very helpful, because it lets us clarify that the present results are agnostic/general: throughout, we do *not* assume realizability. In particular, the bounds are statements about excess risk relative to the best model in $\Theta$ under the true distribution, exactly as in classical agnostic learning. What changes under performativity is *which distribution* the benchmark is taken with respect to, since the deployed model changes the data-generating process. This is the source of the extra terms.
>
> It is also useful to spell out what the **realizable case** would and would not buy us here. Realizability means there exists some $\theta^\star\in\Theta$ with zero or Bayes-optimal risk under the data-generating distribution $d_0$. In our setting, even if such a $\theta^\star$ exists for the initial distribution $d_0$, deployment can change the distribution to $d_1=\mathrm{Tr}(d_0,\theta^\star)$ and then further to $d_2,d_3,\ldots$. So realizability at $d_0$ does *not* eliminate the performative penalty, because the model may be evaluated under a distribution it helped create. In other words, realizability can remove or reduce the “approximation error” part of the problem, but not the “distribution-shift error” generated by performativity.
>
> This also clarifies the role of **properness**. It enters only because the current analysis studies RERM over $\Theta$, i.e., the learner outputs a model in the hypothesis class. But the statistical structure of the bounds is not tied to proper learning. If one instead had an improper learner or an approximate optimizer, the natural extension of our bounds is $\text{statistical term} + \text{performative shift term} + \text{optimization/algorithmic term}$.
>
> $~$
>
> **To sum up**, it appears the main issue you raised was neither about correctness nor relevance of our results. Our theory *does* provide the usual shrinking-with-$n$ behavior including rates, but the draft did not state this clearly enough next to the theorem (changed now, see details above and answer to ThZS). Since this is only a presentation issue that did not require new/adapted results, we would greatly appreciate it if you could reconsider your assessment, particularly whether the paper is deemed acceptable once this presentation issue has been addressed – given your indication that the work is both sound and relevant.
>
> We would, of course, be happy to address any remaining questions you may have. Thank you!

---

> > ### Author Rebuttal · Reviewer_rjij · 2026-04-02
> >
> > I appreciate the response by the authors.  The response did not provide a lot of new information.  The bounds do not follow standard approaches in PAC learning and this sentiment is also shared by Reviewer Ehak even after the rebuttal.  Nevertheless, with the additional clarifications and promises by the reviewers for the final version I am also happy to raise the score from weak reject (3) to weak accept (4).

---

> > > ### Author Response · Authors · 2026-04-03
> > >
> > > Thank you very much for your thoughtful follow-up and for reconsidering our paper in light of the rebuttal. We greatly appreciate your careful reading and your willingness to update your assessment.
> > >
> > > We understand your concern that the bounds do not read as transparently as standard PAC-style results, and that making their interpretation more immediate would strengthen the paper. In the revision, we will therefore focus on improving this aspect of the presentation.
> > >
> > > Thank you again for the constructive feedback and for helping us improve the manuscript.

---

### Official Review · Reviewer_EveS · 2026-03-12

**Soundness:** 3
**Presentation:** 4
**Significance:** 4
**Originality:** 4
**Overall Recommendation:** 5
**Confidence:** 3

**Summary:**

The paper provides the conceptual and technical foundations for performative learning theory, i.e., for the development of generalization bounds when there exist feedback loops between a model's predictions and the distributions of data it aims to predict. The authors (i) formalize several notions of "generalization" in the presence of performative effects, (ii) provide generalization bounds that pertain to each of these multiple notions of generalization, and (iii) empirically illustrate insights from these bounds in a real-world dataset.

**Compliance With Llm Reviewing Policy:**

Affirmed.

**Final Justification:**

The paper provides a thorough conceptual and theoretical analysis that departs substantially from prior work. The authors' rebuttal clarified the questions I raised, and I maintain my recommendation to accept the paper.

**Key Questions For Authors:**

1. Do I correctly understand that your analysis requires that the training data are drawn i.i.d. from the population, i.e., that $\hat{d}_t \overset{iid} \sim d_t$, $\forall t \in$ { $1, \ldots{}, T$ }? If so, could you please comment on this assumption in the context of the motivating examples illustrated in Figure 1, and on how violation of this assumption would affect your analysis and results?
2. I was surprised to see that $s$ does not appear in the bound in Theorem 3.10, especially given the result in Corollary 3.11. Could you please elaborate on the dependence of the performative excess risk on $s$?

**Limitations:**

yes

**Strengths And Weaknesses:**

**Strengths**
The paper provides a thorough conceptual and theoretical analysis of generalization in the presence of performative effects, and is exceptionally clearly written.

The paper's contributions also appear to have the potential for both theoretical and practical significance, and to depart substantially from prior work. Performative effects arise in many settings of practical interest, some of which the authors discuss in detail. The authors carefully taxonomize different notions of generalization for the performative setting in Table 1, and provide novel results pertaining to seven of these notions. I am not familiar with existing theoretical work on performative prediction; in light of the background provided by the authors, these results seem to both build substantially on and/or be more general than existing results.

**Weaknesses**
The i.i.d. assumption seems to preclude the types of settings illustrated in Figure 1, which motivate the significance of the authors' analysis. In these settings, the sample is not drawn randomly from the population, but instead consists of a well-defined subpopulation (e.g., a model learning from the behavior of users in San Francisco aims to generalize to the entire Bay Area).

---

> ### Author Rebuttal · Authors · 2026-03-27
>
> Thank you for taking the time to engage so carefully with our work. We appreciate your encouraging assessment, especially your view that the paper provides a “thorough conceptual and theoretical analysis” and is “exceptionally clearly written.”
>
>
> $~$
>
> **Dependence of the performative excess risk on $s$:** We *estimate* $s \in \[0,1\]$ (an unobserved population quantity) based on our observations of its empirical analogue $\frac{m}{n}$ with $m \leq n$. (See lines 275-279, right column.) This is why $s$ is not in our bounds, but $\frac{m}{n}$ is. If $s$ were in our bounds, they would be underdetermined (and of little practical use due to uncertainty over unknown $s$). It is easy to see that $\frac{m}{n}$ (the share of, say, registered (i.e., in our sample) navigation app users reacting to predictions) is an estimator of the share $s$ of all potential (i.e., in population) navigation app users reacting to predictions. (Refer to proof of Lemma 3.9 for details.) Since our generalization bounds (like the one in mentioned Thm. 3.10) hold with high probability, we have to study the tail behaviour of the distribution of our estimator $\frac{m}{n}$. (Specifically, we do this via Wald’s method, see line 275 (right column).) This is exactly what Lemma 3.9 and Corollary 3.11 achieve: they give a high probability upper bound on unknown $s$ that depends on observed (thus known) $m$ and $n$. We will add this explanation to the camera-ready version.
>
> $~$
>
> With respect to the **i.i.d. assumption**, you asked two questions: (1) Isn’t it violated in Fig. 1? Answer: Yes. (2) Is i.i.d. required in all $t$ for our results to hold? Answer: No.
>
> 1. You are completely correct about (1) and we sincerely thank you for pointing this out. The motivating examples in Figure 1 suggest a source--target / subpopulation mismatch (geographic subpopulation of SF and Bavaria rather than an i.i.d. sample from whole Bay Area and Germany, respectively). This is conceptually distinct from the initial i.i.d. assumption. We revise the illustration in Fig. 1 accordingly, now showing a random sample of car drivers from the Bay area as well as a random sample of job centers in Germany to better align conceptually with the i.i.d. assumption required for the results.
>
> 2. This already hints at our answer to (2): No, our results do *not* assume that the training data is drawn i.i.d. from the population at every round. In fact, this is the setup considered in prior work (namely, Sec. 3.4 in Perdomo et al. (2020)), as indicated by our Table 1. Our theory starts from an initial i.i.d. sample and then analyzes how sample distributions and/or population distributions evolve under performative feedback. For instance, we cover settings with sample performativity (RQ1), where the training sample $\widehat d_t$ itself evolves through the performative shifts, e.g. $ \forall t \in \{1, \dots, T\}: \widehat d_t = \mathrm{Tr}(\widehat d_{t-1}, \widehat\theta_t)$, so there is no fresh resampling and hence no i.i.d. assumption beyond initialization. We now make this distinction much more explicit in the discussion of Figure 1 and the surrounding setup.
>
> $~$
>
> **To sum up**, we thank the reviewer for the helpful feedback, helping us put the finishing touches on the paper. In our reading, the remaining issues were about the i.i.d. assumption and $s$ in Theorem 3.10/Corollary 3.11, which we have addressed in the revision. *Thus, we would greatly appreciate it if you could reconsider whether the paper is now closer to a strong accept.* Otherwise, we would, of course, be happy to address any remaining questions you may have.

---

> > ### Author Rebuttal · Reviewer_EveS · 2026-04-01
> >
> > Thanks to the authors for the clarifying rebuttal, which has fully addressed my concerns. I maintain my recommendation to accept the paper.

---

> > > ### Author Response · Authors · 2026-04-03
> > >
> > > Thank you for the follow-up and for indicating that your concerns have been fully resolved! We are glad the clarification was helpful.
> > >
> > > If you still see some room for improvement, we would of course be very grateful for a brief pointer and would be happy to work towards further improving our paper in that direction.
> > >
> > > In any case, we are grateful for your thoughtful reading and helpful feedback, which really improved the paper. Thank you!

---

### Decision · Program_Chairs · 2026-04-30

**Decision:**

Accept (regular)

**Comment:**

This paper studies the generalizability in performative learning, where both the data distribution and the model change over time and influence each other, by introducing four benchmark metrics to quantify different notions of generalization in such a setting, and deriving upper bounds on these metrics.

There was a robust rebuttal and follow-on discussion between the author(s) and the reviewers, which led two reviewers to increment their scores. Overall, the reviewers found the problem well motivated and are largely positive, although there remained some unresolved issues, such as the strength of the main technical assumption and the tightness of the bounds.